# Spatial and temporal modeling of breast cancer mortality in Kansas: An R-INLA approach

Stephanie Colwell[1], Prabhakar Chalise[1,2], Byron Gajewski[1,2], Isuru Ratnayake[1,2], Dinesh Pal Mudaranthakam[1,2]*

**1** Department of Biostatistics & Data Science, University of Kansas Medical Center, Kansas City, Kansas, United States of America, **2** The University of Kansas Cancer Center, University of Kansas Medical Center, Kansas City, Kansas, United States of America

* dmudaranthakam@kumc.edu

## Abstract

### Introduction

Based on Breast Cancer Statistics, 2025, breast cancer is a leading cause of death among women in the United States. Geographic disparities and associated risk factors influence breast cancer mortality over time and across spatial areas within the state of Kansas.

### Objective

This study investigates the spatial and temporal distribution of breast cancer mortality in Kansas, analyzing associations with socioeconomic, healthcare, and behavioral characteristics while accounting for geographic heterogeneity and temporality.

### Methods

Using data from 105 counties within Kansas, breast cancer mortality was modeled using known count distributions. Within these model frameworks, two approaches to spatial units were implemented: using county-level units and creating spatial clusters of counties. These models incorporated both spatially structured and unstructured effects with different correlation structures. Key socioeconomic, healthcare, and behavioral factors were analyzed. Model performance was evaluated using the Deviance Information Criterion (DIC), Widely Applicable Information Criterion (WAIC), and Marginal Log Likelihood.

### Results

The Poisson BYM2 model provided the best fit for the county analysis (DIC = 1305.02, WAIC = 1308.40) and the spatial cluster analysis (DIC = 2435.90, WAIC = 2420.70). The percent of females who binge drink alcohol was significant in the county analysis. In contrast, the average percent of females who binge drink alcohol, the average

**Data availability statement:** All data used to reach the conclusions of this study are publicly available and accessible without restriction from the sources listed below. The minimal data set underlying the findings including all values used to calculate summary statistics, generate figures, and fit the spatial–temporal models can be fully reconstructed from these sources following the procedures described in the Methods and the accompanying analysis code. Mortality outcome data were obtained from the Kansas Information for Communities (KIC) Health Information Portal: https://kic.kdhe.ks.gov/death_new.php#top Area-level sociodemographic and health-related covariates were obtained from the Kansas Health Matters Portal: https://www.kansashealthmatters.org/ Measures of community assets and rurality were obtained from the Community Assets and Relative Rurality (CARR) Index: https://ruralityindex.com/ All data processing steps, analytic workflows, model inputs, and code used to generate results, tables, and figures are publicly available in a GitHub repository: https://github.com/spepperkumc/Spatial-Temporal-Modeling-INLA The GitHub repository includes a README file with step-by-step instructions describing how to (1) access the raw public data, (2) construct the analytic dataset, (3) reproduce all statistical analyses, and (4) regenerate all figures and reported results, thereby enabling full replication of the study.

**Funding:** This study was supported by National Cancer Institute Cancer Center Support Grant P30CA168524.

**Competing interests:** No Competing Interests.

percent of females who smoke tobacco, the average percentage of females with diabetes, and the average percent of females were significant in the spatial cluster analysis. The relative risk of breast cancer mortality did not change significantly over time in the county analysis, but it did in the cluster analysis.

## Conclusions

Spatial and temporal models provide valuable insights into the risk of breast cancer mortality in Kansas, within the county analysis and the spatial cluster analysis. Public health officials should focus on providing resources and equitable healthcare in high-risk counties and high-risk spatial clusters through targeted interventions to improve access to healthcare and breast cancer outcomes.

## Introduction

In the United States of America, breast cancer is one of the leading causes of death among women [1]. Although breast cancer mortality rates in the United States have been decreasing annually, the rate of this decrease has slowed in recent years [2]. Despite the steady decline, breast cancer still poses a significant threat among women. According to the Kansas Department of Health and Environment, female breast cancer is the top cancer diagnosis among females in Kansas between 2017 and 2021 and was recorded as the second-highest cancer death among females in Kansas between 2019 and 2023 [3]. Within the State of Kansas, the age-adjusted incidence rate of female breast cancer was 135.6 per 100,000 between 2017 and 2021 [3], and the age-adjusted female breast cancer mortality rate in Kansas was 19.4 deaths per 100,000 females between 2019 and 2023 [3]. The age-adjusted incidence rate of female breast cancer was 130.8 per 100,000 women between 2018 and 2022, with an age-adjusted mortality rate of 19.2 per 100,000 annually in the United States between 2019 and 2023 [4].

When attempting to determine breast cancer mortality rates, certain risk factors need to be considered. Numerous studies have highlighted various risk factors for female breast cancer, including screening practices [5], smoking [6], diabetes [7], obesity [7,8], physical inactivity [9], race [9], educational attainment [10], poverty [10], and environmental influences like air pollution [10–12]. Other potential risk factors, such as age at menarche, age at menopause, breastfeeding, parity, and the percentage of the population born abroad, could also be relevant. However, obtaining comprehensive data on these factors for all counties in the contiguous United States may be highly challenging or even unfeasible [13]. Due to the inability to directly capture key risk factors for breast cancer mortality at the county level, researchers can use spatial methods to adjust models for these inherent differences within the counties. According to the 2020 Census [14], 59 of Kansas's 105 counties, or 56%, were classified as rural, with an urban index of less than 0.40. It has been shown that individuals living in rural areas are more likely to experience hardships in healthcare, such as higher travel costs and a greater need to access care due to limited resources in their

area [15]. These burdens can negatively affect breast cancer outcomes, and by accounting for rurality, public health efforts can better tailor interventions, allocate resources more equitably, and develop policies that improve access to timely and high-quality care.

Spatial and temporal modeling of breast cancer mortality, especially within small spatial areas, is not without its challenges. It is typical for cancer registries and state health agencies to have specific criteria related to the release of data for small geographic areas. A standard practice within small area estimation is to report cases that are greater than 10 [16]. Small area estimation of breast cancer mortality, such as within counties in Kansas, can present challenges when the number of breast cancer mortality in certain counties is very low, often less than 10 cases. These small counts not only lead to statistical instability of spatial and temporal models but also raise serious concerns about patient privacy and confidentiality, as individuals could be potentially identified, especially in sparsely populated areas [16]. To address this issue, spatial clustering techniques can be applied to group neighboring counties with similar characteristics, thereby forming spatial clusters that each contain more than 10 breast cancer mortality cases. This approach can improve the reliability of statistical estimates while also protecting individual privacy by aggregating data to a level that maintains confidentiality.

A similar Bayesian spatial–temporal modeling framework using R-INLA was previously applied by Deblina Khana et al. (2018) to estimate mortality rates across U.S. counties at the national level. In contrast, the present study focuses on University of Kansas Cancer Center Catchment area, with the goal of producing locally actionable small-area estimates to inform regional cancer control and resource allocation.

In this study, we estimate county-level spatiotemporal variation in breast cancer mortality using Bayesian hierarchical models fitted via Integrated Nested Laplace Approximation (INLA). To accommodate overdispersion, excess zeros, and spatial–temporal dependence, we compare Poisson (Po), generalized Poisson (GP), negative binomial (NB), and their zero-inflated counterparts (ZIP, ZINB) under Besag-York-Mollié (BYM), Besag-York-Mollié 2 (BYM2) for space, and Random Walk order 1 (RW1) and order 2 (RW2) for time, including county-specific effects and spatial clustering. Models adjust for socioeconomic, healthcare access, and lifestyle covariates, and we generate maps of relative risk over space and time. We select the specification that best balances fit and interpretability using DIC, WAIC, and Marginal Likelihood, to characterize how mortality risk evolves geographically and temporally after accounting for known risk factors. By incorporating spatial mapping, we identify hotspots, areas with persistently elevated risk based on Bayesian exceedance probability mapping, providing actionable locations for targeted resource allocation and funding to help reduce breast cancer mortality. By incorporating spatial mapping of breast cancer mortality, we can identify hotspots, defined as areas most in need of action. This will give healthcare providers and policy makers key areas that are in most need and can receive better resource allocation and funding to assist in reducing their breast cancer mortality.

## Methods

### Data sources

This study used data on breast cancer deaths in 105 counties in the state of Kansas over a four-year period from 2018 to 2021. The data were obtained from the Kansas Information for Communities (KIC) health information portal [17]. This data uses breast cancer deaths from malignant neoplasms of the breast, explicitly between 2018 and 2021. In addition to mortality counts, the study incorporates various demographic, healthcare, and behavioral characteristics. These covariates were obtained from the Kansas Health Matters portal [18]. The county level covariates that are incorporated are percentage of female [18], number of primary care physicians per 100,000 population [18], percentage of females who binge drink [18], percentage of females who smoke cigarettes [18], percentage of females over age 20 that are obese, percentage of females over age 20 that have been diagnosed with diabetes [18], and the community assets and relative rurality (CARR) index [19]. This is the latest curated data available and accessible for mortality counts. Data collection details for each covariate are described below.

The percentage of females is defined as the percentage of the population that is female. This covariate was captured through the U.S. Census Bureau Population and Housing Unit Estimates. The number of primary care physicians is defined as the primary care provider rate per 100,000 population [18]. Primary care providers included any practicing physician who specialized in general practice medicine, family medicine, internal medicine, or pediatrics. This covariate was captured through County Health Rankings, which is maintained by the Conduent Healthy Communities Institute [18]. The percentage of females who binge drink is defined as the percentage of female adults who reported binge drinking at least once during the 30 days prior to the survey. Female binge drinking is defined as drinking four or more drinks on one occasion. This covariate was captured by CDC – PLACES and is maintained by Conduent Healthy Communities Institute [18]. The percentage of females who smoke is defined as the percentage of female adults who currently smoke cigarettes. This covariate was captured by CDC-PLACES and is maintained by Conduent Healthy Communities Institute [18]. The percentage of females who are obese is defined as the percentage of female adults aged 20 and older who are obese according to the Body Mass Index (BMI). The BMI is calculated by taking the females weight and dividing it by their height squared in metric units. A BMI greater than or equal to 30 is considered to be obese. This covariate was captured by the Centers for Disease Control and Prevention and maintained by Conduent Healthy Communities Institute [18]. The percentage of females over 20 with diabetes is defined as the percentage of females aged 20 and older who have ever been diagnosed with diabetes. Females who were diagnosed with diabetes only during the course of their pregnancy were not included. This covariate was captured by the Centers for Disease Control and Prevention and maintained by Conduent Healthy Communities Institute [18]. The community assets and relative rurality (CARR) index is defined as a continuous, multi-dimensional measure of rurality based on the concept of sustainable development that integrates measures of environmental, social, and economic resources. The CARR index varies between 0 and 1 where values closer to 0 represent urban areas and values closer to 1 represent rural areas [19]. The CARR values in Kansas ranged from 0.178 to 0.740, reflecting meaningful variation in rurality across counties. However, the majority of counties were clustered within a moderate range (median = 0.396; IQR: 0.315–0.566), suggesting that extreme rurality differences are less common. All data were accessed for research purposes on July 25th, 2025, and authors did not have access to information that could identify individual participants during or after data collection.

Expected counts were calculated using internal (crude) standardization rather than age-standardized reference populations due to limited availability and instability of age-specific county-year data, particularly in counties with small counts. Accordingly, estimated relative risks reflect comparative spatial patterns rather than fully age-adjusted mortality rates.

In the following section, we will discuss the spatial-temporal Bernardinelli model that is implemented on the breast cancer mortality count data and covariates described in this section. It should be noted that even though the data used are breast mortality counts, the model below uses the population of each county as an offset in order to model the relative risk of breast cancer mortality adjusted for the differing population sizes within each county in Kansas.

## Spatial-temporal Bernardinelli model

This paper implements the Bernardinelli spatial-temporal model. Let $i = 1, 2, \ldots, n$ denote the spatial units and $j = 1, 2, \ldots, T$ denote the temporal units. The Bernardinelli [20] model assumes that the observed number of cases, $Y_{ij}$, in spatial unit $i$ at temporal unit $j$ follows a Poisson distribution:

$$Y_{ij} \sim Po\left(\eta_{ij}\right),$$

where the Poisson mean (i.e., $E\left(Y_{ij}\right) = \eta_{ij}$) is defined as:

$$\eta_{ij} = E_{ij}\Theta_{ij}.$$

Taking the logarithm of the mean yields the log-linear formulation:

$$\log(\eta_{ij}) = \log(E_{ij}) + \log(\Theta_{ij}),$$

where $\log(E_{ij})$ is included as an offset term to account for differences in population size across spatial units and temporal units.

Here, $E_{ij}$ denotes the expected number of cases, and $\Theta_{ij}$ represents the relative risk for the spatial unit $i$ at temporal unit $j$. The expected counts were calculated using internal standardization based on the overall crude incidence rate across all spatial units at each temporal unit:

$$E_{ij} = N_{ij} \times \frac{\sum_{k=1}^{n} Y_{kj}}{\sum_{k=1}^{n} N_{kj}},$$

where $N_{ij}$ is the population at risk at spatial unit $i$ at temporal unit $j$. Thus, relative risk, $\Theta_{ij}$ represents the ratio of the risk in spatial unit $i$ at temporal unit $j$ to the overall average risk across all spatial units at the same temporal unit, as defined by the internally standardized reference rate used to compute $E_{ij}$. Values greater than 1 (i.e., $\Theta_{ij} > 1$) indicate higher-than-average risk, while values less than 1 (i.e., $\Theta_{ij} < 1$) indicate lower-than-average risk. Equivalently, $\Theta_{ij} = 1$ implies that the risk in that spatial unit and time period is equal to the overall average risk at that time.

The relative risk is modeled on the log scale as:

$$\log(\Theta_{ij}) = \alpha + u_i + v_i + (\beta + \delta_i) \times t_j \tag{1}$$

where $\alpha$ denotes the intercept, $u_i$ and $v_i$ represent spatial random effect, $\beta$ is a global linear trend effect, and $\delta_i$ represents spatial deviations from the global temporal trends. Area-specific covariates can be incorporated by replacing the intercept parameter $\alpha$ in equation (1) with $X_{ij}\Phi$, where $X_{ij}$ denotes the covariates for spatial unit $i$ at time $j$ and $\Phi$ represents the regression coefficients. It is noteworthy to mention that the reported relative risks are obtained from the fitted spatio-temporal model and are adjusted for included covariates as well as spatial and temporal random effects.

## Temporal specification

Across all spatial–temporal models, $t_j$ denotes the temporal index (year). Because the study period includes only four years (2018–2021), temporal patterns are inherently weakly identifiable, and inference regarding long-term trends is constrained. Accordingly, a parsimonious temporal specification was adapted in which time enters linearly through $t_j$ under the Bernardinelli formulation. The coefficients $\beta$ captures the overall year-to-year effect, while $\delta_i$ allows for area-specific deviations from global temporal trend. Weakly informative Gaussian priors were assigned to global temporal coefficient $\beta$ to stabilize estimation.

Differences observed between county-level and cluster-level analyses reflect aggregation effects, as clustering increases case counts and reduces variability, making short-term differences more detectable. Temporal findings should therefore be interpreted cautiously.

## Spatial correlation structures

Several spatial dependence structures are incorporated through alternative prior specifications for the structured spatial effect $u_i$ in equation (1). In this study, spatial dependence is modeled by using the Besag-York-Mollie (BYM), Besag-York-Mollie 2 (BYM2), Random Walk of Order 1 (RW1), and Random Walk of Order 2 (RW2) models.

The BYM model incorporates an Intrinsic Conditional Autoregressive (ICAR) model for spatial dependence and a random effect for non-spatial heterogeneity. The ICAR model assumes that $u_i$ follows a Conditional Autoregressive (CAR) model such that:

$$u_i | u_{-i} \sim N\left(\overline{u}_{\delta_i}, \frac{\sigma_u^2}{n_{\delta_i}}\right),$$

(2)

where $\overline{u}_{\delta_i} = n_{\delta_i}^{-1} \sum_{j \in \delta_i} u_j$. In (2), $\delta_i$ represents the set of neighbors to the spatial unit $i$ and $n_{\delta_i}$ represents the number of neighbors of the spatial unit $i$. The ICAR model smooths the spatial effect according to a neighborhood structure in which two areas are considered neighbors if they share a common boundary. The unstructured spatial random effects are assumed independent:

$$v_i \sim N\left(0, \sigma_v^2\right).$$

(3)

Under the BYM specification, the spatial–temporal linear predictor in equation (1) remains unchanged, with spatial dependence entering through the prior distributions of $u_i$ and $v_i$.

The BYM2 model is a reparametrized version of the BYM model [21], introducing the mixing parameter $\varphi \in [0, 1]$ and the precision parameter $\tau_\gamma$. The combined spatial effect is written as:

$$u_i^* + v_i^* = \sqrt{\frac{1}{\tau_\gamma}}\left(\sqrt{\varphi} \cdot u_i + \sqrt{1-\varphi} \cdot v_i\right).$$

This formulation improves identifiability and stabilizes the partitioning of spatial variance [22].

The key difference between the BYM and BYM2 models is in how they structure and scale the random effects $u_i$ and $v_i$. The BYM2 model provides a more consistent decomposition of spatial variance, making it easier for Markov Chain Monte Carlo (MCMC) convergence.

The RW1 prior assumes:

$$u_1 \sim N\left(0, \sigma_u^2\right),$$

$$u_i | u_{i-1} \sim N\left(u_{i-1}, \sigma_u^2\right) \quad i = 2, \ldots, n.$$

(4)

Here, $i - 1$ refers to the preceding county in the defined geographic ordering. This specification captures smooth large-scale spatial gradients rather than adjacency-based local dependence.

The Random Walk of Order 2 prior penalizes second-order differences,

$$\Delta^2 u_i = u_i - 2u_{i-1} + u_{i-2} \sim N\left(0, \sigma_u^2\right),$$

$$u_i | u_{i-1}, u_{i-2} \sim N\left(2u_{i-1} - u_{i-2}, \sigma_u^2\right) \quad i = 3, \ldots, n,$$

(5)

Encouraging a smoother quadratic spatial trend across the ordered counties.

The BYM and BYM2 frameworks decompose spatial variation into structured dependence and unstructured random heterogeneity, with BYM2 employing a reparameterization that improves identifiability and stabilizes variance partitioning across spatial components. The RW1 and RW2 specifications provide alternative spatial smoothing structures by penalizing first- and second-order differences (equations 4 and 5), respectively, across ordered spatial units. Within the INLA framework, weakly informative priors were specified for hyperparameters to ensure model stability while minimizing undue prior influence on posterior estimates. These assumptions guide how each model captures spatial dependence and uncertainty, providing a clear rationale for comparing model specifications.

The spatial-temporal framework described in equation (1) can be adjusted to incorporate different count distributions such as Zero-Inflated Poisson, Negative Binomial, Zero-Inflated Negative Binomial, and Generalized Poisson.

In the following section, we will discuss the integrated nested Laplace approximation method for Bayesian estimation of the Bayesian spatial-temporal hierarchical models discussed in this section.

## Integrated Nested Laplace Approximation (INLA)

The Integrated Nested Laplace Approximation (INLA) is an approximate Bayesian inference method that provides a computationally efficient alternative to Markov Chain Monte Carlo (MCMC) for latent Gaussian models [23]. INLA is specifically designed for models that can be expressed as latent Gaussian Markov random fields (GMRFs), a class that includes spatial, temporal, and spatial–temporal hierarchical models. Latent Gaussian models amenable to INLA can be expressed through three hierarchical layers: the likelihood, the latent Gaussian field, and the hyperparameters. The general formulation is:

$$\boldsymbol{y}|\boldsymbol{\xi}, \psi_1 \sim \prod_i p\left(y_i|\xi_i,\ \psi_1\right) \tag{6}$$

$$\boldsymbol{\xi}|\psi_2 \sim p\left(\boldsymbol{\xi}|\psi_2\right) = N\left(\boldsymbol{\mu}\left(\psi_2\right),\ \boldsymbol{Q}^{-1}\left(\psi_2\right)\right) \tag{7}$$

$$\psi = \left(\psi_1,\ \psi_2\right) \sim p(\psi) \tag{8}$$

where $\boldsymbol{y} = \{y_i\}_{i=1,2\ldots,n}$ denotes the observed data, $\boldsymbol{\xi} = \{\xi\}_{i=1,2,\ldots,n}$ denotes the latent Gaussian field, and $\psi = (\psi_1,\ \psi_2)$ denotes the vector of hyperparameters. The matrix $Q\left(\psi_2\right)$ is the precision (inverse covariance) matrix governing the Gaussian prior of the latent field.

In the context of the Bernardinelli spatial–temporal model described in equations (3)–(5), the latent Gaussian field $\boldsymbol{\xi}$ corresponds to the full linear predictor for all spatial–temporal units. Specifically,

$$\xi_{ij} =\ X_{ij}\Phi_{ij} + u_i +\ v_i +\ \left(\beta + \delta_i\right) \times t_j,$$

so that the spatial random effects $(u_i, v_i)$, temporal components, and space–time interaction terms form elements of the latent field. The hyperparameters $\psi$ include all precision parameters governing the spatial ICAR effects, unstructured spatial effects, temporal random walk components, and any mixing parameters those in the BYM2 specification. Thus, the Bernardinelli model in equations (1)–(3) is a specific instance of the general latent Gaussian model class estimated using INLA.

Although spatial dependence is incorporated in the model, it enters through the prior distribution of the latent Gaussian field (equation 7) rather than through the likelihood. Conditional on the latent field $\boldsymbol{\xi}$ and hyperparameters $\psi$, the

observations $Y_{ij}$ are assumed independent, Therefore, the likelihood retains the product form given in equation 6, and the spatial dependence is encoded is the precision matrix $Q(\psi_2)$.

The relative risk $\Theta_{i,j}$ is a deterministic transformation of the linear predictor and defined as:

$$\Theta_{ij} = \exp(\xi_{ij}),$$

and therefore, does not receive an independent prior. Its posterior distribution is induced by the posterior distribution of the latent field and hyperparameters.

The joint posterior distribution of the latent Gaussian model is denoted as:

$$p(\xi, \psi|\boldsymbol{y}) \propto p(\psi)p(\xi|\psi)\prod_i p(y_i|\xi_i, \psi_1).$$

Here, $p(\xi|\psi)$ corresponds directly to the Gaussian prior specified in the equation (7), and $p(\psi)$ is defined in equation 8.

Inference proceeds by integrating over the latent field and hyperparameters to obtain marginal posterior distributions. In particular, hyperparameter marginals are obtained by integrating out $\xi$,

$$p(\psi_k|\boldsymbol{y}) = \int p(\xi, \psi|\boldsymbol{y})\, d\xi,$$

while marginal posterior distributions for components of the latent field are obtained by integrating over the hyperparameters,

$$p(\xi_i|\boldsymbol{y}) = \int p(\xi_i, \psi|\boldsymbol{y})\, p(\psi|\boldsymbol{y})\, d\psi.$$

These integrals are approximated numerically using nested Laplace approximations within the INLA framework [23,24]. Since the dimension of the latent field $\xi$ can be very large ($10^2 - 10^5$), while the number of hyperparameters $\psi$ is relatively small (typically 2–5, but not to exceed 20). This structured approximation enables efficient computation of marginal posterior distributions. The INLA algorithm is implemented using the R-INLA package [23].

## Hotspot definition

In this study, hotspot areas were identified using exceedance probabilities, defined as the posterior probability that the area-specific relative risk (RR) exceeds a prespecified threshold $c$, i.e., $\Pr(RR > c|data)$. Exceedance probability mapping is commonly used in Bayesian disease mapping to highlight areas with unusually elevated risk while accounting for the posterior uncertainty [25].

For the primary hotspot definition, this study used $c = 1.5$, a moderate elevation above the base line risk, and an area was classified as a hotpot if $\Pr(RR_{it} > 1.5) > 0.75$. A relative-risk threshold of 1.5 has been used in prior exceedance mapping to represent a practically meaningful elevation in risk beyond background variability [26].

To evaluate robustness to threshold choice, a sensitivity analysis was conducted by repeating hotspot identification under alternative criteria, $RR > 2.0$ with probability >0.80 (more stringent) and $RR > 1.5$ with probability >0.70 (less stringent). These alternatives were selected to examine whether hotspot locations remain consistent when requiring stronger risk elevation and/or greater posterior certainty.

In the following section, we will discuss the Spatial Kluster Analysis by Tree Edge Removal (SKATER) algorithm that will be implemented to generate spatial clusters of counties within the state of Kansas. This algorithm is used to generate new spatial units that can be used in small area estimation analysis to ensure patient privacy and confidentiality.

### Spatial Kluster analysis by Tree Edge Removal (SKATER)

Spatial clusters were generated using the SKATER algorithm [27], an established regionalization method for partitioning spatial units into contiguous and internally homogeneous clusters. The implementation in this study does not introduce methodological novelty; rather, SKATER was applied to address instability arising from small-area counts. The algorithm operates on a spatial connectivity graph (Fig 1), where counties are represented as vertices and edges represent spatial adjacency. Edges were defined using first-order queen contiguity, meaning that counties sharing either a common boundary or a common vertex were considered neighbors.

While adjacency defines the neighborhood structure, clustering requires distinguishing between more and less similar neighboring counties. Therefore, edges were assigned weights based on multivariate dissimilarity between standardized county-level covariates using Euclidean distance [24]. The existence of an edge alone indicates spatial contiguity but does not capture similarity in demographic, behavioral, or healthcare characteristics; thus, adjacency by itself is insufficient for identifying meaningful regional partitions.

SKATER constructs a minimum spanning tree (MST) from the weighted connectivity graph and iteratively removes edges with high dissimilarity to partition the graph into spatially contiguous clusters that minimize within-cluster variance [27–29]. The MST ensures that all counties remain connected prior to partitioning and provides a systematic framework for identifying spatially contiguous and internally homogeneous clusters. During the clustering procedure, a constraint can be imposed on a specific attribute, such as the population within a spatial cluster or the number of disease cases within a spatial cluster. The SKATER algorithm was implemented as described above in the spatial cluster analysis.

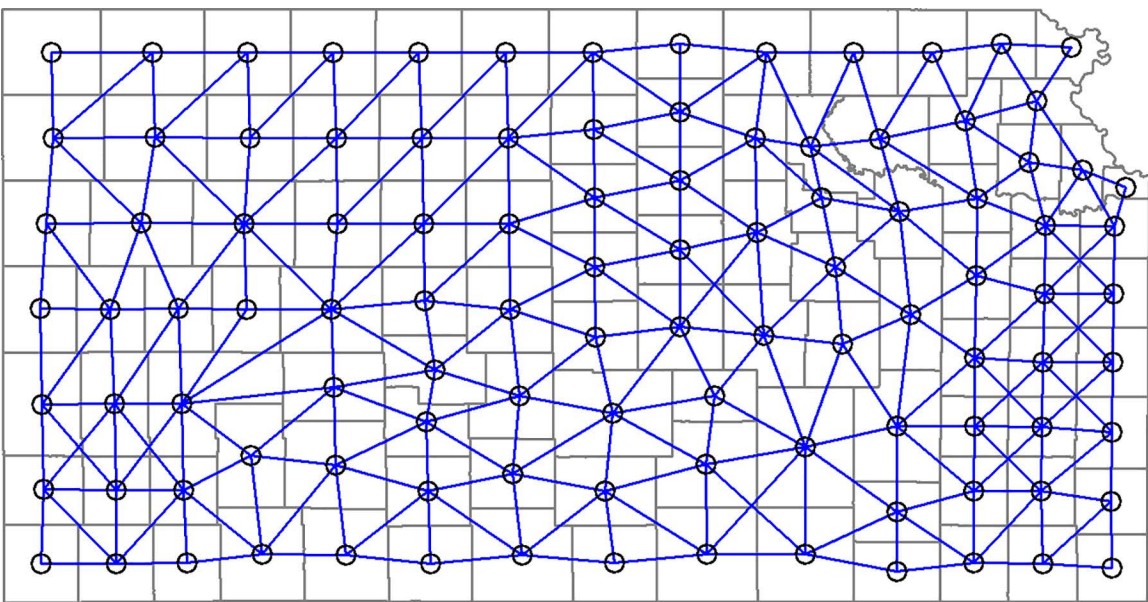

**Fig 1. Connectivity graph used in the SKATER algorithm.**

## Ethics approval

This study was reviewed and approved by the University of Kansas Medical Center Institutional Review Board and was determined to be non–human subject's research.

## Results

### Descriptive statistics

Table 1 shows the range of values for breast cancer mortality in Kansas from 2018–2021, along with socio-demographic and behavioral characteristics that are associated with breast cancer mortality. Noticeable between-county variation in breast cancer mortality was observed each year, with counts ranging from 0–88 in 2018, 0–67 in 2019, 0–68 in 2020, and 0–67 in 2021. The average breast cancer mortality is constant across years, with an average breast cancer mortality of 3.55 in 2018, 3.90 in 2019, 3.28 in 2020, and 3.59 in 2021. Many counties had breast cancer mortality cases below the mean due to a few extreme values. Among the socio-demographic and behavioral characteristics that are known to increase the risk of cancer, female smoking and diabetes prevalence showed slight decreasing patterns from 2018–2021,

**Table 1. County-Level Summary Statistics for Breast Cancer Mortality Cases and Their Associated Variables in Kansas between 2018-2021.**

| Variable | Year | Mean (SD) | Median (IQR) | Minimum | Maximum |
|---|---|---|---|---|---|
| **Cases** | 2018 | 3.55 (10.50) | 1.00 (3.00) | 0 | 88 |
| | 2019 | 3.90 (9.47) | 1.00 (4.00) | 0 | 67 |
| | 2020 | 3.28 (9.11) | 1.00 (3.00) | 0 | 68 |
| | 2021 | 3.59 (8.92) | 1.00 (4.00) | 0 | 67 |
| **Population (per 10,000)** | 2018 | 2.77 (7.95) | 0.69 (1.69) | 0.12 | 59.8 |
| | 2019 | 2.77 (8.00) | 0.69 (1.68) | 0.12 | 60.2 |
| | 2020 | 2.80 (8.12) | 0.69 (1.71) | 0.13 | 61.1 |
| | 2021 | 2.80 (8.14) | 0.70 (1.68) | 0.13 | 61.5 |
| **Female (%)** | 2018 | 49.70 (1.36) | 50.00 (1.00) | 43.5 | 52.1 |
| | 2019 | 49.70 (1.36) | 49.90 (1.10) | 43.3 | 51.6 |
| | 2020 | 49.20 (1.35) | 49.50 (1.30) | 42.7 | 51.3 |
| | 2021 | 49.20 (1.41) | 49.40 (1.10) | 42.3 | 51.3 |
| **Obese (%)** | 2018 | 28.90 (4.84) | 28.70 (6.50) | 17.7 | 40.7 |
| | 2019 | 29.40 (4.79) | 29.20 (7.30) | 21.8 | 41.2 |
| | 2020 | 28.80 (5.84) | 28.20 (8.40) | 17.3 | 43.3 |
| | 2021 | 30.40 (5.70) | 29.90 (7.80) | 19.1 | 42.8 |
| **Diabetes (%)** | 2018 | 7.96 (1.39) | 7.70 (1.70) | 5.5 | 12.7 |
| | 2019 | 7.90 (1.18) | 7.70 (1.60) | 5.8 | 12.3 |
| | 2020 | 7.74 (1.36) | 7.50 (1.80) | 5.2 | 11.9 |
| | 2021 | 7.75 (1.32) | 7.50 (1.50) | 5.5 | 13.5 |
| **Smoking (%)** | 2018 | 18.60 (1.74) | 18.40 (2.40) | 12.5 | 23.3 |
| | 2019 | 17.00 (1.49) | 16.90 (2.00) | 11.8 | 20.3 |
| | 2020 | 17.70 (1.68) | 17.60 (2.10) | 11.1 | 23.2 |
| | 2021 | 17.20 (1.76) | 17.20 (2.10) | 10 | 22 |
| **Binge Drinking (%)** | 2018 | 15.10 (1.25) | 15.00 (1.20) | 12.5 | 19.3 |
| | 2019 | 15.90 (1.35) | 15.70 (16.0) | 13 | 20.9 |
| | 2020 | 16.00 (1.36) | 15.70 (1.30) | 13.6 | 21.9 |
| | 2021 | 16.60 (1.36) | 16.30 (1.50) | 14 | 23.7 |

whereas binge drinking and obesity demonstrated modest increases over time. The Bayesian-estimated annual changes were small ($\beta_{smoking}$ = −0.35, 95% CI: −1.31 to 0.61; $\beta_{binge\ drinking}$ = 0.45, 95% CI: −0.09 to 0.96; $\beta_{obesity}$ = 0.40, 95% CI: −0.56 to 1.38; $\beta_{diabetes}$ = −0.08, 95% CI: −0.24 to 0.08 percentage points per year), and all 95% credible intervals included zero, indicating no statistically significant temporal trends during the study period. It is noteworthy to mention that Female (%) represents the proportion of the county population that is female based on census data.

Fig 2 presents descriptive histograms of county-level breast cancer case counts from 2018 to 2021, illustrating temporal variation in disease burden. It can be observed that most counties report fewer than 10 breast cancer mortality cases each year, with most counties reporting zero breast cancer mortality cases. Few counties report having breast cancer mortality cases larger than 20, with the largest breast cancer mortality cases of 88 in 2018.

## Kansas County model comparisons

Table 2 summarizes Bayesian model comparison metrics used to evaluate competing specifications, including the Deviance Information Criterion (DIC), the Widely Applicable Information Criterion (WAIC), and the marginal log-likelihood. Lower DIC and WAIC values indicate improved expected out-of-sample fit after accounting for model complexity. Several candidate models demonstrated comparable fit, with relatively small differences in DIC and WAIC across the leading specifications. The Poisson BYM2 model was used as the primary model for inference because it yielded the most favorable combination of fit and parsimony among the leading specifications (DIC = 1305.02; WAIC = 1308.40), while acknowledging that several alternative specifications provided comparable fit.

It can be seen that the DIC and WAIC values for Poisson, Generalized Poisson, and Negative Binomial distributions with the BYM and BYM2 spatial correlation structures are relatively close, suggesting that there is no clear, overwhelming superior model in terms of fit and complexity balance. In Bayesian model comparison, small differences in DIC are

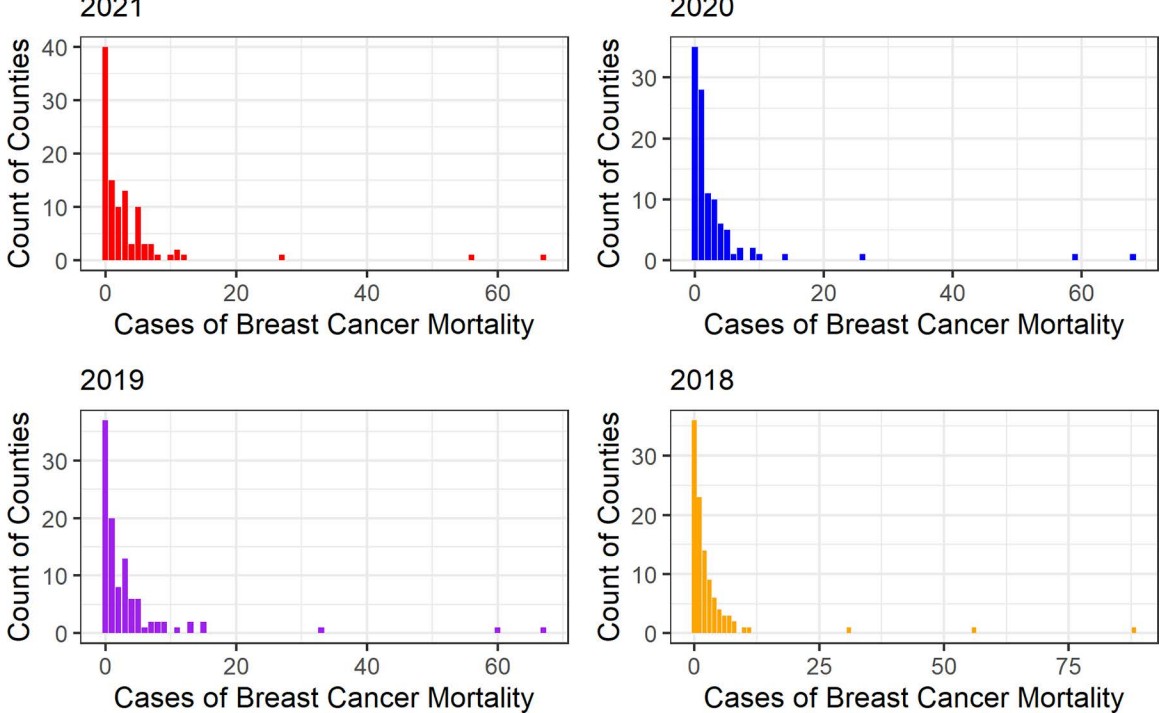

**Fig 2. Variation in Breast Cancer Mortality Across Kansas Counties (2018-2021).**

**Table 2. Model Fit Statistics for Spatial Temporal Models using R-INLA.**

| Distribution | Correlation Structure | DIC | WAIC | Marginal Log Likelihood |
|---|---|---|---|---|
| **Poisson** | BYM | 1309.35 | 1313.49 | −704.89 |
| | BYM2 | **1305.02** | **1308.4** | **−655.91** |
| | RW1 | 1314.21 | 1316.54 | −729.12 |
| | RW2 | 1316.58 | 1320.71 | −735.12 |
| **ZI-Poisson (0)** | BYM | 1489.35 | 1489.76 | −794.58 |
| | BYM2 | 1483.55 | 1485.35 | −746.24 |
| | RW1 | 1494.17 | 1495.7 | −819.52 |
| | RW2 | 1497.85 | 1500.6 | −826.61 |
| **ZI-Poisson (1)** | BYM | 1311.82 | 1315.14 | −707.74 |
| | BYM2 | 1307.53 | 1310.09 | −658.83 |
| | RW1 | 1316.34 | 1318.24 | −732.03 |
| | RW2 | 1318.46 | 1321.76 | −738.06 |
| **Generalized Poisson** | BYM | 1310.77 | 1309.43 | −656.5 |
| | BYM2 | 1310.77 | 1309.9 | −656.95 |
| | RW1 | 1330.11 | 1329.3 | −729.21 |
| | RW2 | 1327.9 | 1328.21 | −733.13 |
| **Negative Binomial** | BYM | 1313.41 | 1315.37 | −705.47 |
| | BYM2 | 1308.93 | 1309.9 | −656.56 |
| | RW1 | 1318.74 | 1318.89 | −729.59 |
| | RW2 | 1321.87 | 1322.82 | −735.47 |
| **ZI-Negative Binomial (0)** | BYM | 1491.54 | 1491.63 | −795.01 |
| | BYM2 | 1487.74 | 1487.23 | −746.81 |
| | RW1 | 1500.57 | 1499.36 | −819.8 |
| | RW2 | 1504.79 | 1504.92 | −826.85 |
| **ZI-Negative Binomial (1)** | BYM | 1321.86 | 1321.1 | −710.05 |
| | BYM2 | 1310.55 | 1311.41 | −659.49 |
| | RW1 | 1321.43 | 1320.99 | −732.5 |
| | RW2 | 1324.09 | 1325.02 | −738.41 |

generally considered negligible and may indicate that the models perform similarly with respect to the observed data. Taking this into account, the final model was a spatiotemporal Poisson model with a BYM2 spatial structure—not only due to its lowest DIC and WAIC and highest marginal log-likelihood, but also for its parsimony, interpretability, and computational efficiency relative to more complex alternatives. Given the large number of zero counts, we also evaluated zero-inflated Poisson and zero-inflated negative binomial alternatives; however, these models did not show a clear improvement in fit compared with the leading non–zero-inflated specifications (Table 2), suggesting that spatial smoothing and alternative dispersion structures captured most of the observed variability.

## Kansas County final model

Table 3 presents the posterior estimates of fixed effects under the BYM2 spatial correlation structure assuming a Poisson distribution. Percentage female binge drinking did not show increased risk in this population ($\beta$ = −0.078; 95% CI: −0.146, −0.006). Although the estimated coefficient is counter intuitively negative, its magnitude is small and the upper bound of the credible interval is very close to zero. This direction of association is counterintuitive given established individual-level evidence linking alcohol consumption to increased breast cancer risk. The observed minimal negative

 

**Table 3. Posterior Estimates and 95% Credible Intervals of Fixed Effects using the BYM2 correlation structure with Poisson Distribution.**

| Coefficient | Estimate (95% CI) |
|---|---|
| Intercept | 3.735 (−1.183, 8.611) |
| Rurality Index (CARR) | 0.001 (−0.013, 0.017) |
| Percent Female | −0.033 (−0.113, 0.047) |
| Percent Female Diabetes | −0.046 (−0.122, 0.031) |
| Percent Female Obese | −0.007 (−0.028, 0.014) |
| PCP | −0.001 (−0.002, 0.001) |
| Percent Female Smoke | −0.008 (−0.061, 0.043) |
| Percent Female Alcohol | −0.078 (−0.146, −0.006) |
| Time | 0.020 (−0.041, 0.080) |

association at the ecological level likely reflects aggregation effects, collinearity with rurality and related behavioral covariates, or residual spatial confounding rather than a protective effect. Pairwise correlations among covariates are presented in Supplementary S4 Fig, where moderate correlations are observed among several behavioral and rurality-related predictors, supporting the possibility of multicollinearity within the multivariable model. It is noteworthy to mention that all covariates were retained in the model because they represent key factors related to breast cancer risk, behavioral characteristics, and healthcare access examined in this study. Other covariates including rurality index, percent female, percent female with diabetes, percent female with obesity, primary care physicians (PCP), and percent female who smoke had 95% credible intervals that included zero but were retained in the model due to their established relevance in breast cancer epidemiology. The absence of statistical significance for several established covariates does not imply model inadequacy, but may reflect spatial smoothing, aggregation effects inherent to county-level data, and shared variance among correlated predictors. The estimated global time effect (trend) in the model was small ($\beta$ = 0.020; 95% CI: −0.041, 0.080) and its 95% credible interval included zero, indicating no statistically significant linear year-to-year trend in breast cancer mortality risk at the county level.

Fig 3 presents the estimated relative risk of breast cancer mortality for all Kansas counties between 2018 and 2021 (Fig 3a), as well as the 95% credible intervals (Fig 3b, 3c). In Fig 3a, each panel represents the relative risk estimates for breast cancer mortality, quantifying whether the breast cancer mortality risk in a county is higher (RR > 1) or lower (RR < 1) than the average risk in Kansas during that year. Relative risk values that are greater than 1 are shown in red and relative risk values that are less than 1 are shown in blue. The relative risk patterns appear broadly similar across years, with no marked visual shifts in the spatial distribution of elevated or reduced risk. Fig 3b shows the lower bound of the 95% credible interval for the relative risk of breast cancer mortality and Fig 3c shows the upper bound of the 95% credible interval for the relative risk of breast cancer mortality. Table 4 presents the five counties with the strongest evidence that the relative risk differs from 1 (null hypothesis $H_0 : \Theta_{ij} = 1$), defined as counties whose posterior 95% credible interval for $\Theta_{ij} = \exp(\xi_{ij})$ excluded 1. In 2018, 11 counties had 95% credible intervals that were above the value of one, and two counties had 95% credible intervals that were less than the value of one. Table 4 shows that Chautauqua, Cowley, Greenwood, Woodson, and Kingman counties had higher relative risk of breast cancer mortality compared to the average relative risk of breast cancer mortality in Kansas, and Wyandotte and Douglas counties had 95% credible intervals had a lower relative risk of breast cancer mortality compared to the average relative risk of breast cancer mortality in Kansas during 2018. In 2019, Chautauqua, Woodson, Greenwood, Ottawa, and Labette counties had a higher relative risk of breast cancer mortality, and Wyandotte, Douglas, and Johnson counties had a lower relative risk of breast cancer compared to the average relative risk of breast cancer mortality in Kansas. In 2020, Chautauqua, Greenwood, Cowley, Woodson, and Kingman counties had a higher relative risk of breast cancer mortality, and Wyandotte, Douglas, Riley, and

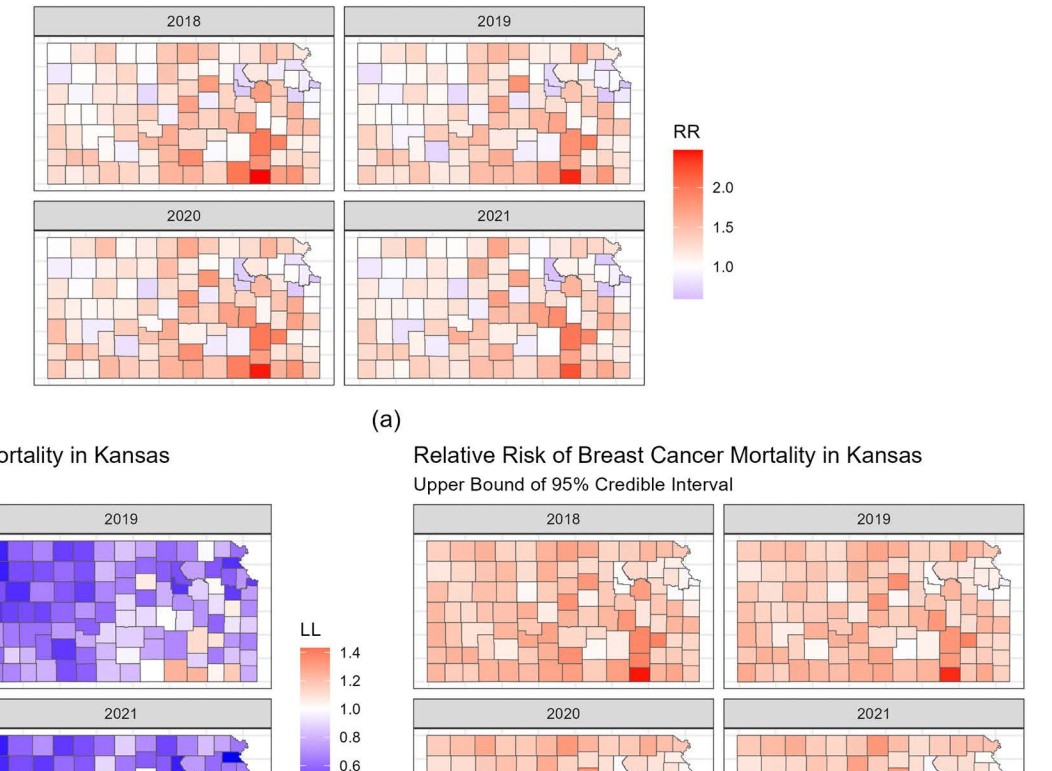

## Fig 3. Relative Risk and 95% Credible Interval of Breast Cancer Mortality for Kansas Counties between 2018 and 2021.

Sedgwick counties had a lower relative risk of breast cancer compared to the average relative risk of breast cancer mortality in Kansas. In 2021, Chautauqua, Greenwood, Woodson, Marion, and Labette counties had a higher relative risk of breast cancer mortality, and Wyandotte, Riley, and Douglas counties had a lower relative risk of breast cancer compared to the average relative risk of breast cancer mortality in Kansas. The counties that had the highest relative risk of breast cancer mortality are counties with a more rural demographic, while the counties that had the lowest relative risk of breast cancer mortality are counties with a more urban demographic. The rurality index (CARR) values for the counties with high relative risk of breast cancer mortality are between 0.32 and 0.39 and the rurality index (CARR) values for the counties with low relative risk of breast cancer mortality are between 0.17 and 0.26.

In order to determine hotspots, or counties that have unusual elevation in relative risk of breast cancer mortality, we can calculate the probability of the relative risk estimates being greater than a given threshold, called exceedance probabilities, for each county. These probabilities are useful in assessing unusual elevation in breast cancer mortality risk. In this study, $RR > 1.5$ was used to represent a moderate evaluation in risk (at least 50% higher than the statewide average for that year). Counties were classified as hotspots when the exceedance probability exceeded 0.75, indicating high posterior support for elevated risk while avoiding labeling areas with substantial posterior uncertainty as hotspots. Fig 4 shows

**Table 4. Top 5 Counties with high or low relative risk compared to the average relative risk in Kansas.**

| Year | County | Relative Risk (95% CI) | Year | County | Relative Risk (95% CI) |
|------|--------|------------------------|------|--------|------------------------|
| 2018 | Chautauqua | 2.47 (1.27, 4.47) | 2018 | Wyandotte | 0.67 (0.51, 0.88) |
|      | Cowley | 2.00 (1.43, 2.72) |      | Douglas | 0.72 (0.53, 0.94) |
|      | Greenwood | 1.98 (1.20, 3.17) |      |  |  |
|      | Woodson | 1.92 (1.04, 3.24) |      |  |  |
|      | Kingman | 1.85 (1.11, 3.01) |      |  |  |
| 2019 | Chautauqua | 2.41 (1.24, 4.35) | 2019 | Wyandotte | 0.63 (0.48, 0.81) |
|      | Woodson | 1.95 (1.09, 3.23) |      | Douglas | 0.65 (0.48, 0.86) |
|      | Greenwood | 1.82 (1.11, 2.90) |      | Johnson | 0.85 (0.74, 0.97) |
|      | Ottawa | 1.81 (1.07, 2.99) |      |  |  |
|      | Labette | 1.77 (1.17, 2.59) |      |  |  |
| 2020 | Chautauqua | 2.44 (1.19, 4.53) | 2020 | Wyandotte | 0.65 (0.48, 0.85) |
|      | Greenwood | 2.00 (1.22, 3.17) |      | Douglas | 0.68 (0.50, 0.88) |
|      | Cowley | 1.96 (1.42, 2.65) |      | Riley | 0.69 (0.48, 0.96) |
|      | Woodson | 1.95 (1.07, 3.30) |      | Sedgwick | 0.87 (0.76, 0.99) |
|      | Kingman | 1.83 (1.10, 2.96) |      |  |  |
| 2021 | Chautauqua | 2.21 (1.14, 3.98) | 2021 | Wyandotte | 0.59 (0.44, 0.78) |
|      | Greenwood | 2.00 (1.21, 3.21) |      | Riley | 0.61 (0.41, 0.88) |
|      | Woodson | 1.88 (1.00, 3.24) |      | Douglas | 0.65 (0.48, 0.86) |
|      | Marion | 1.71 (1.09, 2.60) |      |  |  |
|      | Labette | 1.65 (1.08, 2.43) |      |  |  |

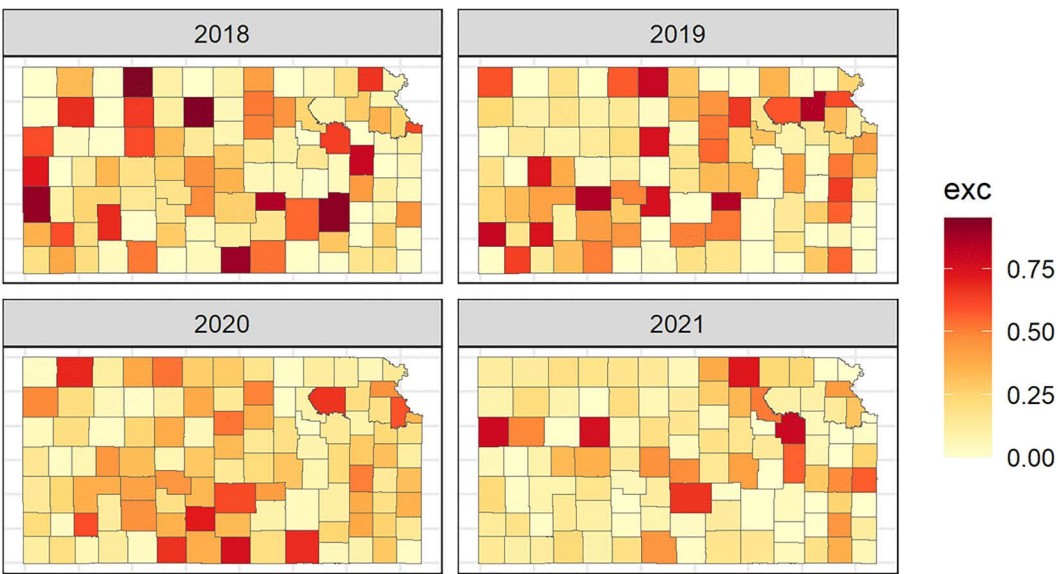

**Fig 4. Map of Exceedance Probabilities in Kansas Counties (>1.5).**

the exceedance probabilities within Kansas counties from 2018 to 2021. Sensitivity analyses were conducted using the alternative hotspot definition (RR > 2.0, with probability>0.80, and RR > 1.50 with probability >0.70); results are reported in Supplementary Material S5 File.

This map provides evidence of excess risk of breast cancer mortality within individual counties from 2018 to 2021. Counties with probabilities close to 1 are very likely to have relative risk that exceeds 1.5, counties with probabilities close to 0 are very unlikely to have relative risk that exceeds 1.5, and counties with probabilities around 0.50 have the highest uncertainty and correspond to having relative risk below or above 1.5 with equal probability. Fig 4 can be used to determine hotspot areas by identifying counties that have large exceedance probabilities. The larger the exceedance probability the darker red the county and the smaller the exceedance probability the more yellow the county. A county would be considered a hotspot for breast cancer mortality if the exceedance probability is greater than 0.75. Table 5 gives the counties that are classified as hotspots. In 2018 and 2019, there were 7 counties identified as hotspots for elevated relative risk of breast cancer mortality. The number of hotspots decreased to one county identified as a hotspot in 2020, and then the number of hotspots increased to 3 in 2021. The hotspot locations did not remain constant over time, with some counties exhibiting elevated exceedance probabilities in certain years and attenuating in others. This temporal variability suggests that geographic risk is dynamic and may reflect shifts in demographic composition, healthcare access, or behavioral risk factors such as smoking and alcohol use. These findings highlight the importance of continued spatial surveillance rather than reliance on static geographic risk classifications. The counties identified as hotspots are known to have a more rural demographic and population.

Sensitivity analyses showed that the spatial pattern of hotspot identification was generally consistent under alternative exceedance thresholds, with fewer hotspots detected under the more stringent criterion (RR > 2.0; probability >0.80) and slightly more hotspots under the less stringent criterion (RR > 1.5, probability >0.70). Detailed county-level sensitivity hotspot listings and maps are provided in the Supplementary Materials S5 File.

A major challenge we encountered during this analysis was the challenge of counties reporting small counts of breast cancer mortality. Small-area counts can significantly affect spatial modeling by introducing high variability and instability in estimates. For example, many counties reported zero breast cancer mortality cases in a given year, while others reported substantially higher counts (e.g., up to 88 cases in 2018), resulting in highly skewed distributions and variance exceeding

**Table 5. Kansas Hotspot Counties Identified by Marginal Exceedance Probability.**

| Year | County | Exceedance Probability |
|---|---|---|
| 2018 | Norton | 0.952 |
| | Osborne | 0.944 |
| | Greenwood | 0.922 |
| | Hamilton | 0.908 |
| | Harper | 0.895 |
| | Harvey | 0.857 |
| | Osage | 0.795 |
| 2019 | Hodgeman | 0.856 |
| | Jackson | 0.853 |
| | Harvey | 0.847 |
| | Stanton | 0.808 |
| | Smith | 0.801 |
| | Stafford | 0.761 |
| | Russell | 0.755 |
| 2020 | Harper | 0.763 |
| 2021 | Wabaunsee | 0.793 |
| | Wallace | 0.786 |
| | Trego | 0.766 |

the mean. To address this instability and reduce the influence of extreme small-area fluctuations, a spatial cluster analysis was implemented.

## Cluster model comparisons

To address the limitations that many counties had small breast cancer mortality counts in the county-level analysis, we implemented a spatial clustering approach to generate larger continuous spatial units that would eliminate the small-count issue. We implemented a constrained spatial regionalization clustering algorithm known as SKATER to cluster counties into a new contiguous spatial clusters. Spatial contiguity was defined using first-order queen adjacency, whereby counties sharing either a common boundary or vertex were considered neighbors. Clustering was implemented in R using the *skater* function within the *spdep* package. We generated spatial clusters using only the 2018 county data and assumed these spatial clusters would remain constant across all time points to preserve spatial coherence and avoid introducing additional temporal clustering variability. Temporally aggregating data across multiple years prior to clustering could mask small-area heterogeneity and dilute the small-count structure that motivated the clustering approach.

The clustering variables were the percent of women who are obese, the percent of women who have diabetes, the percent of females, the percent of females who smoke, the percent of females who binge drink alcohol, the number of primary care physicians, and the rurality index (CARR) were used to determine the edge costs associated with each edge in the spatial neighbor list. All covariates were standardized into z-scores, with mean of 0 and standard deviation of 1. Dissimilarity between each county and its neighboring counties was computed based on these standardized variables to generate contiguous spatial clusters. Euclidean distance was used to measure dissimilarity within the SKATER algorithm. Each spatial cluster was required to include at least 10 breast cancer mortality cases to further reduce disclosure risk and ensure confidentiality in reporting. To ensure that all spatial clusters had at least 10 breast cancer mortality cases in all years, the 2018 clusters were constrained to include more than 20 cases.

Fig 5 shows an elbow curve of the within-cluster sums of squares for different cluster values. When deciding the optimal number of clusters, the elbow, or the point at which the within-cluster sums of squares begin to flatten, identifies the point at which additional clusters yield diminishing reductions in within-cluster variability. In this case, the rate of decrease in within-cluster sums of squares becomes progressively smaller after six clusters, indicating diminishing returns in additional cluster partitioning. Therefore, six was selected as the optimal number of spatial clusters, balancing within-cluster homogeneity and between-cluster separation while satisfying the minimum case constraint. Fig 6 shows the new spatial

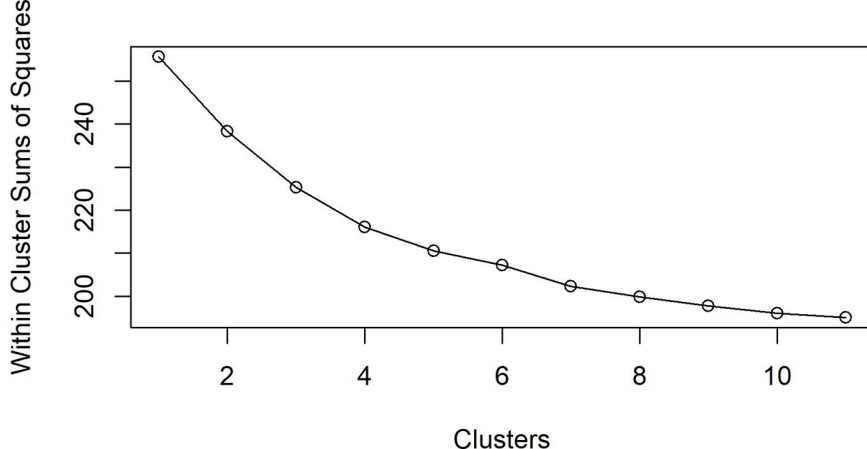

**Fig 5. Elbow Curve for SKATER Clustering.**

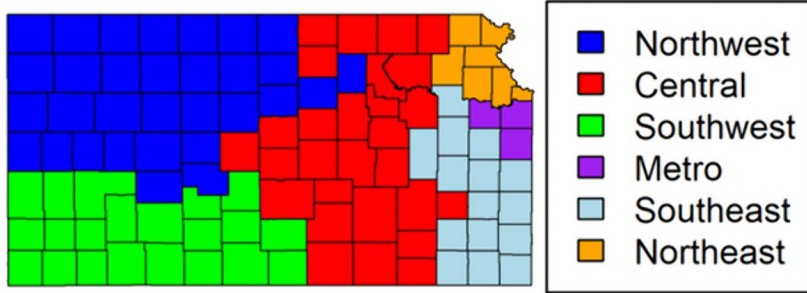

**Fig 6. New Spatial Units Identified Through Cluster Analysis using SKATER.**

clusters generated from 105 counties in Kansas. A complete listing of county-to-cluster assignments is provided in Supplementary S3 Table to ensure reproducibility. The six spatial clusters were named based on their location within the state. The six spatial clusters are the Northwest, Southwest, Central, Northeast, Southeast, and Metro. These new spatial clusters will be used within the spatial-temporal modeling discussed in the county analysis.

Table 6 presents a summary of key Bayesian fit indices used to evaluate and compare the performance of the estimated models for the cluster analysis. This table includes the Deviance Information Criterion (DIC), the Widely Applicable Information Criterion (WAIC), and the Marginal Log Likelihood. These criteria help assess model quality by balancing model fit and complexity; lower values of DIC and WAIC indicate better-fitting models. Based on the model fit indices in Table 6, several candidate specifications provided comparable fit, with only modest differences in DIC and WAIC across the leading models. The Poisson BYM2 specification achieved the lowest DIC and WAIC among the Poisson-based cluster models (DIC = 2435.90; WAIC = 2420.70). We therefore selected this model as the primary cluster-level specification for inference due to its favorable fit, parsimony, and interpretability, while acknowledging that competing models demonstrated similar performance.

It can be seen that the DIC and WAIC values for all probability distributions and spatial correlation structures are relatively close, suggesting that there is no clear, superior model in terms of fit and complexity balance. In Bayesian model comparison, slight differences in DIC are considered negligible and may indicate that the models perform similarly with respect to the observed data. Taking this into account, the final model was chosen to be the cluster spatial-temporal model, assuming a Poisson distribution with the BYM2 spatial correlation structure due to its simplicity.

## Cluster final model analysis

Table 7 presents the posterior estimates of fixed effects under the BYM2 spatial correlation structure assuming a Poisson distribution. In Table 7, average percent of females who binge drink alcohol (95% CI: −0.91, −0.28), average percent of females who smoke tobacco (95% CI: −0.28, −0.05), average percentage of females with diabetes (95% CI: −0.58, −0.02), and average percent female (95% CI: 0.93, 1.57) were found to be statistically significant. Although all other covariates of average rurality index and average percentage of females who are obese are not found to be statistically significant, they are kept in the model due to them being known risk factors for breast cancer mortality. The cluster-level model suggested short-term year-to-year differences in relative risk over the study period.

Differences in statistical significance between the county-level and cluster-level analyses likely reflect aggregation effects and variance reduction. By combining counties into larger spatial units, total case counts increase and variability decreases, improving statistical stability and power to detect associations. However, aggregation may also introduce ecological bias; therefore, cluster-level findings should be interpreted within the ecological framework of the study.

**Table 6. Model Fit Statistics for Spatial Temporal Cluster Models using R-INLA.**

| Distribution | Correlation Structure | DIC | WAIC | Marginal Log Likelihood |
|---|---|---|---|---|
| **Poisson** | BYM | 2436.06 | 2420.80 | −1309.89 |
| | BYM2 | **2435.90** | **2420.70** | **−1304.01** |
| | RW1 | 2435.90 | 2420.58 | −1316.24 |
| | RW2 | 2435.84 | 2420.48 | −1316.68 |
| **ZI-Poisson (0)** | BYM | 2439.05 | 2423.30 | −1314.33 |
| | BYM2 | 2438.81 | 2423.15 | −1308.44 |
| | RW1 | 2439.33 | 2423.64 | −1319.49 |
| | RW2 | 2439.28 | 2423.90 | −1317.78 |
| **ZI-Poisson (1)** | BYM | 2439.05 | 2423.30 | −1314.33 |
| | BYM2 | 2438.81 | 2423.15 | −1308.44 |
| | RW1 | 2439.33 | 2423.64 | −1319.50 |
| | RW2 | 2439.28 | 2423.91 | −1317.78 |
| **Generalized Poisson** | BYM | 2439.09 | 2423.33 | −1316.57 |
| | BYM2 | 2438.96 | 2423.29 | −1309.98 |
| | RW1 | 2439.27 | 2423.33 | −1322.20 |
| | RW2 | 2439.19 | 2423.21 | −1322.63 |
| **Negative Binomial** | BYM | 2438.05 | 2422.30 | −1312.94 |
| | BYM2 | 2437.82 | 2422.17 | −1307.06 |
| | RW1 | 2438.37 | 2422.66 | −1318.11 |
| | RW2 | 2438.32 | 2422.91 | −1316.39 |
| **ZI-Negative Binomial (0)** | BYM | 2440.61 | 2424.59 | −1317.38 |
| | BYM2 | 2440.28 | 2424.40 | −1311.50 |
| | RW1 | 2441.12 | 2425.01 | −1322.55 |
| | RW2 | 2441.16 | 2425.34 | −1320.83 |
| **ZI-Negative Binomial (1)** | BYM | 2440.61 | 2424.58 | −1317.38 |
| | BYM2 | 2440.27 | 2424.40 | −1311.50 |
| | RW1 | 2440.71 | 2424.50 | −1322.72 |
| | RW2 | 2440.98 | 2424.74 | −1324.07 |

**Table 7. Posterior Estimates and 95% Credible Intervals of Fixed Effects using the BYM2 correlation structure with Poisson Distribution.**

| Coefficient | Estimate (95% CI) |
|---|---|
| Intercept | −48.97 (−66.24, −31.73) |
| Average Rurality Index (CARR) | 0.02 (−0.06, 0.08) |
| Average Percent Female | 1.25 (0.93, 1.57) |
| Average Percent Female Diabetes | −0.30 (−0.58, −0.02) |
| Average Percent Female Obese | −0.004 (−0.06, 0.05) |
| Average Percent Female Smoke | −0.16 (−0.28, −0.05) |
| Average Percent Female Alcohol | −0.59 (−0.91, −0.28) |
| Time | 0.35 (0.17, 0.53) |

Fig 7 presents the estimated relative risk of breast cancer mortality for all six spatial clusters between 2018 and 2021 (Fig 7a), as well as the 95% credible intervals (Fig 7b, 7c). In Fig 7a, each panel represents the relative risk estimates for breast cancer mortality, quantifying whether the breast cancer mortality risk in a cluster is higher (RR > 1) or lower (RR < 1) than the average risk in Kansas during that year. Relative risk values that are greater than 1 are shown in red and relative risk values that are less than 1 are shown in blue. We can see that the relative risk of breast cancer mortality in the Northwest cluster of Kansas stays constant from 2018 to 2019 and then declines, while the relative risk of breast cancer mortality in the Southwest cluster of Kansas increases from 2018 to 2021. The relative risk of breast cancer mortality in the Central cluster and Southeast cluster of Kansas fluctuate up and down from 2018 to 2021 while the Northeast and Metro clusters relative risk of breast cancer stays constant from 2018 to 2021. Fig 7b shows the lower bound of the 95% credible interval for the relative risk of breast cancer mortality, and Fig 7c shows the upper bound of the 95% credible interval for the relative risk of breast cancer mortality. Table 8 presents spatial clusters whose posterior 95% credible intervals for $\Theta_{ij}$ excluded 1. Table 8 shows that the Metro and Northeast spatial clusters had a lower relative risk of breast cancer mortality compared to the average risk of breast cancer mortality in Kansas from 2018 to 2021. The Southwest and the Central clusters had a higher relative risk of breast cancer mortality compared to the average risk of breast cancer in Kansas from

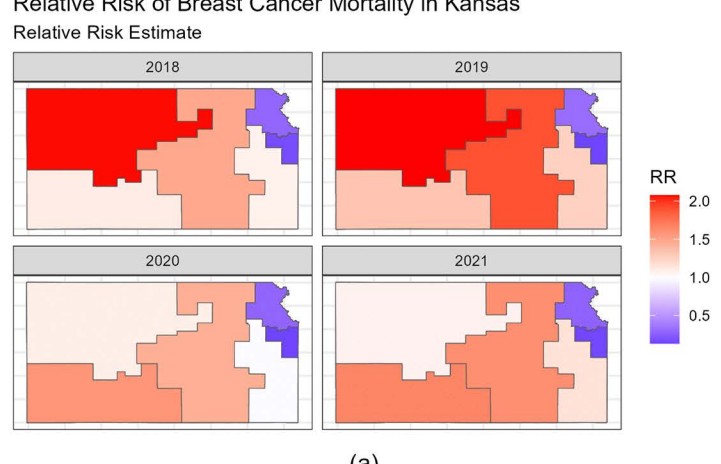

(a)

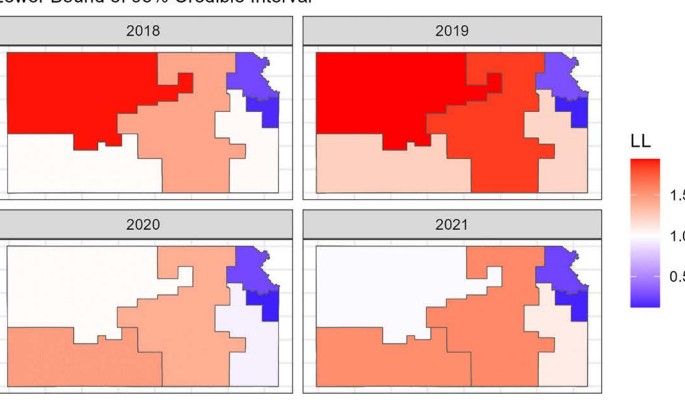

(b)

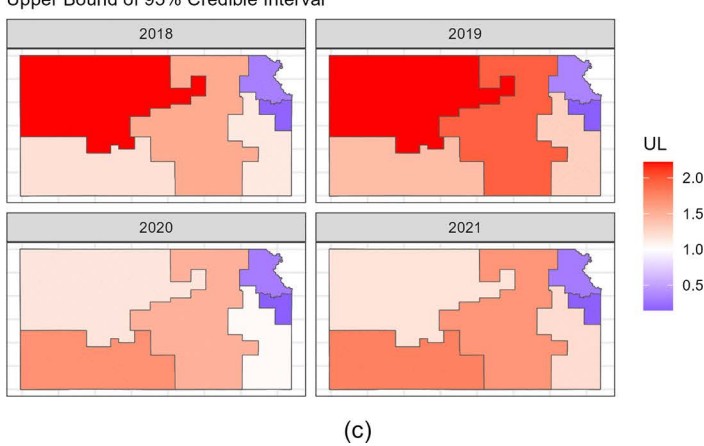

(c)

**Fig 7. Relative Risk and 95% Credible Interval of Breast Cancer Mortality for Kansas Clusters between 2018 and 2021.**

**Table 8. Relative Risk and 95% CI for Kansas Spatial Clusters.**

| Year | Cluster | Relative Risk (95% CI) | Year | Cluster | Relative Risk (95% CI) |
|------|---------|------------------------|------|---------|------------------------|
| 2018 | Northwest | 2.07 (1.93, 2.22) | 2018 | Metro | 0.17 (0.15, 0.18) |
|  | Central | 1.48 (1.44, 1.53) |  | Northeast | 0.29 (0.25, 0.32) |
|  | Southwest | 1.11 (1.02, 1.20) |  |  |  |
|  | Southeast | 1.08 (1.02, 1.14) |  |  |  |
|  |  |  |  |  |  |
| 2019 | Northwest | 2.08 (1.94, 2.22) | 2019 | Metro | 0.14 (0.12, 0.15) |
|  | Central | 1.90 (1.85, 1.95) |  | Northeast | 0.32 (0.29, 0.35) |
|  | Southwest | 1.33 (1.24, 1.42) |  |  |  |
|  | Southeast | 1.25 (1.21, 1.30) |  |  |  |
|  |  |  |  |  |  |
| 2020 | Southwest | 1.58 (1.48, 1.69) | 2020 | Metro | 0.13 (0.12, 0.15) |
|  | Central | 1.45 (1.40, 1.49) |  | Northeast | 0.28 (0.26, 0.31) |
|  | Northwest | 1.09 (1.02, 1.16) |  |  |  |
| 2021 | Southwest | 1.66 (1.55, 1.78) | 2021 | Metro | 0.14 (0.13, 0.16) |
|  | Central | 1.61 (1.57, 1.66) |  | Northeast | 0.27 (0.24, 0.31) |
|  | Southeast | 1.16 (1.10, 1.23) |  |  |  |
|  |  |  |  |  |  |

2018 to 2021. The Southeast cluster had a higher relative risk of breast cancer mortality compared to the average risk of breast cancer mortality in Kansas from 2018 to 2019 and in 2021, and the Northwest cluster had a higher relative risk of breast cancer mortality compared to the average risk of breast cancer mortality in Kansas from 2018 to 2020. The spatial clusters that had the highest relative risk of breast cancer mortality are spatial clusters with a more rural demographic, while the spatial clusters that had the lowest relative risk of breast cancer morality are spatial clusters with a more urban demographic. The average rurality index (CARR) values for the spatial clusters with high relative risk of breast cancer mortality are between 0.31 and 0.56 and the average rurality index (CARR) values for the spatial clusters with low relative risk of breast cancer mortality are between 0.22 and 0.26.

In order to determine hotspots, or clusters that have unusual elevation in relative risk of breast cancer mortality, we can calculate the probability of the relative risk estimates being greater than a given threshold, called exceedance probabilities, for each cluster. These probabilities are useful in assessing unusual elevation in breast cancer mortality risk. In this study, hotspots were defined using the same exceedance probability rule as in the county analysis: RR > 1.5 to represent moderate excess risk and exceedance probability>0.75 to indicate high posterior support. Sensitivity analyses using alternative thresholds (RR > 2.0, probability>0.80; and RR > 1.5, probability>0.70) are provided in Supplementary Material S5 File. Fig 8 shows the exceedance probabilities within Kansas clusters from 2018 to 2021.

This map provides evidence of excess risk of breast cancer mortality within clusters across from 2018 to 2021. Clusters with probabilities close to 1 are very likely to have relative risk that exceeds 1.5, clusters with probabilities close to 0 are very unlikely to have relative risk that exceeds 1.5, and clusters with probabilities around 0.50 have the highest uncertainty and correspond to having relative risk below or above 1.5 with equal probability. Fig 8 can be used to determine hotspot areas by identifying clusters that have large exceedance probabilities. The larger the exceedance probability the darker red the cluster and the smaller the exceedance probability the more yellow the cluster. A cluster would be considered a hotspot for breast cancer mortality if the exceedance probability is greater than 0.75. Table 9 gives the clusters that are classified as hotspots. In 2018, the Northwest cluster was identified as a hotspot for elevated relative risk of breast cancer mortality. In 2019, the Northwest and Central clusters were identified as hotspots. In 2020, the Southwest cluster was

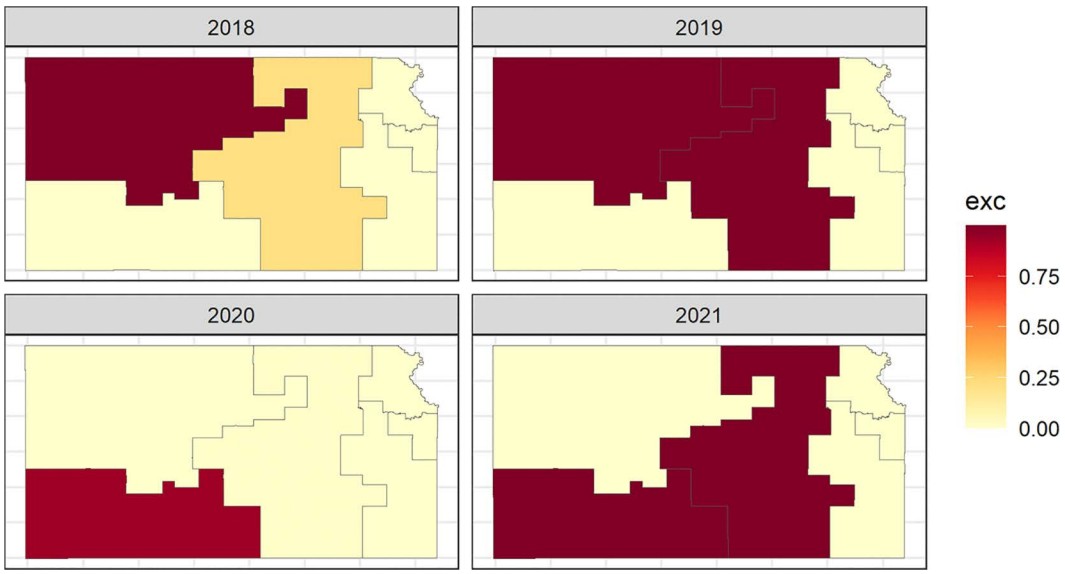

**Fig 8. Map of Exceedance Probabilities in Kansas Clusters.**

**Table 9. Kansas Hotspot Clusters Identified by Marginal Exceedance Probability.**

| Year | Cluster | Exceedance Probability |
|------|---------|------------------------|
| 2018 | Northwest | 1.000 |
| 2019 | Northwest | 1.000 |
| | Central | 1.000 |
| 2020 | Southwest | 0.931 |
| 2021 | Central | 1.000 |
| | Southwest | 0.997 |

identified as a hotspot, and in 2021 the Central and Southwest clusters were identified as hotspots. The clusters identified as hotspots are known to have a more rural demographic and population.

Under alternative exceedance thresholds (RR > 2.0, probability >0.80; RR > 1.5, probability >0.70), hotspot identification showed similar geographic patterns, with stricter thresholds yielding a smaller subset of hotspots and relaxed thresholds yielding modest expansion. Full results are provided in Supplementary Material S5 File.

## Discussion

There are limited studies that have examined the direct and indirect risk factors for breast cancer mortality while simultaneously accounting for spatial and temporal variations. In this study, we examined socioeconomic, healthcare, and behavioral characteristics of breast cancer mortality across different spatial units in the state of Kansas. We implemented spatial and temporal modeling across county spatial units and within spatial clusters of counties generated by the SKATER spatial clustering algorithm. The spatial clustering algorithm was implemented to generate spatial clusters of counties due to a large number of counties reporting breast cancer mortality cases less than 10. We wanted to demonstrate a technique for generating spatial units that preserve patient privacy and confidentiality while enabling spatial and temporal modeling.

In the county-level analysis, percent female binge drinking was statistically significant but exhibited a negative association with breast cancer mortality after adjustment for spatial structure and other covariates. Although extensive individual-level evidence demonstrates that alcohol consumption increases breast cancer risk [30–35], the negative association observed in our ecological model likely reflects aggregation effects, urban–rural structure, correlated behavioral covariates, or residual spatial confounding rather than a protective effect. This discrepancy highlights the distinction between individual-level risk and ecological associations, and the findings should not be interpreted causally.

The Poisson BYM2 model identified several counties in the southeast region of the state with elevated relative risk of breast cancer mortality, corresponding primarily to more rural areas. Although we report results from the Poisson BYM2 model due to its parsimony and interpretability, model comparison results indicated that alternative count specifications (including negative binomial, generalized Poisson, and zero-inflated models) demonstrated similar fit. This suggests that the main conclusions are not driven by a single distributional assumption.

The counties that had the highest relative risk of breast cancer mortality are counties with a more rural demographic, while the counties that had the lowest relative risk of breast cancer mortality are counties with a more urban demographic. The rurality index (CARR) values for the counties with high relative risk of breast cancer mortality are between 0.32 and 0.39, and the rurality index (CARR) values for the counties with low relative risk of breast cancer mortality are between 0.17 and 0.26. This aligns with prior evidence that rural residents experience delayed diagnoses, lower screening rates, and reduced access to oncology services, resulting in poorer outcomes [36]. Although obesity and diabetes are biologically linked to breast cancer progression [37], we did not find them to be significant predictors in the county-level analysis. Certain counties were identified as hotspots for high relative risk of breast cancer mortality through the use of marginal exceedance probabilities, with the specific counties detailed in the Results section. The counties identified as hotspots were more likely to be rural. The number of counties classified as hotspots varied across the study period but showed an overall decline relative to the earlier years.

In the spatial cluster analysis, findings likely reflect broader demographic and behavioral patterns at the population level rather than direct individual effects. Predictors that were not statistically significant may have limited independent signal after adjustment for correlated covariates or reduced variability across counties. Although clustering increases statistical stability by combining counties and increasing case counts, the analysis remains ecological and does not allow for individual-level causal conclusions. The Poisson BYM2 model for the spatial cluster analysis identified clusters with a high relative risk of breast cancer mortality and clusters with fluctuating relative risk over time. The spatial clusters that had the highest relative risk of breast cancer mortality are spatial clusters with a more rural demographic, while the spatial clusters that had the lowest relative risk of breast cancer morality are spatial clusters with a more urban demographic. The average rurality index (CARR) values for the spatial clusters with high relative risk of breast cancer mortality are between 0.31 and 0.56 and the average rurality index (CARR) values for the spatial clusters with low relative risk of breast cancer mortality are between 0.22 and 0.26. We were able to identify certain clusters as hotspots for high relative risk of breast cancer mortality through the use of marginal exceedance probabilities. It was shown that the clusters identified as hotspots were more likely to have a rural demographic.

Temporal fluctuations in relative risk at the county level may reflect demographic shifts, variability in access to screening and treatment services, and statistical instability in sparsely populated areas, even after Bayesian smoothing. Persistent hotspot patterns likely represent structural and healthcare access disparities shared across neighboring counties. These findings are ecological and should be interpreted as hypothesis-generating rather than causal.

The temporal dimension includes a limited number of annual observations, which constrains the identifiability of sustained temporal trends. Accordingly, the temporal component should be interpreted as capturing short-term year-to-year variation rather than long-term change. Differences between the spatial-unit and cluster analyses likely reflect differences in statistical stability and aggregation; clustering increases case counts and reduces variance, which can make short-term temporal differences more detectable.

Overall, this study demonstrates that spatial and temporal modeling provides great insight into the trend of breast cancer mortality by capturing the complex interplay among spatial units and known risk factors. While further research is needed in order to incorporate specific prevention strategies, our findings provide valuable insights into potential hotspots, areas of high risk, and temporal trends of breast cancer mortality in Kansas.

## Limitations

This study has several limitations that should be considered when interpreting the findings. First, retrospective observational design inherently limits causal inference, as it relies on existing data that may be subject to confounding and bias. Additionally, the use of aggregated county-level covariates introduces the risk of ecological fallacy, whereby associations observed at the group level may not hold at the individual level. This analysis did not account for a comprehensive set of known risk factors for breast cancer mortality, such as genetic predispositions and other biological or lifestyle-related variables, due to the inability to capture such data at the county level. The omission of these covariates may influence the interpretation of the findings. Furthermore, counties with small numbers of breast cancer mortality cases may produce unstable estimates and reduced statistical power, even with Bayesian smoothing. Spatial clusters were identified using baseline data and assumed to remain constant over time, which may not fully capture potential shifts in spatial configuration. Finally, the limited number of temporal observations constrains the ability to distinguish between random fluctuation and systematic temporal change; therefore, temporal findings should be interpreted cautiously and not as evidence of sustained long-term trends.

Because this analysis relies on aggregated spatial units and Bayesian spatial smoothing, the estimated relative risks represent stabilized population-level patterns rather than individual-level risk. Spatial smoothing and clustering improve estimate stability in areas with small counts but may attenuate extreme values or mask localized heterogeneity. As a result, identified areas of elevated risk should be interpreted as regions warranting further investigation rather than definitive evidence of localized causal risk.

## Conclusions

This study demonstrates the value of spatial-temporal modeling and cluster analysis in analyzing the relative risk of breast cancer mortality across Kansas. Within the county and spatial cluster analysis, significant geographic variation was identified, emphasizing the importance of incorporating both structured and unstructured heterogeneity. Among all models considered, the Poisson BYM2 model provided the best fit for the county analysis and the spatial cluster analysis. The findings show the importance of addressing socioeconomic, healthcare, and behavioral characteristics that are known risk factors for breast cancer mortality. Public health officials and researchers should implement strategies that prioritize allocating resources and improving breast cancer mortality outcomes in counties and spatial clusters of high-risk to achieve better health outcomes. Implementing spatial and temporal models as described in this paper can assist public health researchers, community members, and policymakers in assessing local breast cancer mortality risks to inform targeted interventions.

Future work will expand this framework by incorporating longer time series to better characterize sustained temporal trends and by exploring finer spatial resolutions or multilevel spatial models, subject to data availability and confidentiality constraints. Extending these methods to multi-state or regional datasets may further enhance understanding of geographic disparities in breast cancer mortality

## Supporting information

**S1 File. Descriptive statistics of all covariates within each county in Kansas over 2018–2021: Mean (Standard Deviation).**
(DOCX)

**S2 Fig. Variation in Average Demographics Across Kansas Counties (2018–2021).**
(DOCX)

**S3 Table. Cluster Membership Table.**
(DOCX)

**S4 Fig. Correlation Matrix.**
(DOCX)

**S5 File. Sensitivity analysis for hotspot thresholds.**
(DOCX)

## Author contributions

**Conceptualization:** Prabhakar Chalise, Byron Gajewski, Isuru Ratnayake, Dinesh Pal Mudaranthakam.

**Data curation:** Stephanie Colwell, Prabhakar Chalise, Isuru Ratnayake.

**Formal analysis:** Stephanie Colwell, Prabhakar Chalise, Byron Gajewski, Isuru Ratnayake, Dinesh Pal Mudaranthakam.

**Investigation:** Stephanie Colwell, Dinesh Pal Mudaranthakam.

**Methodology:** Stephanie Colwell, Prabhakar Chalise, Byron Gajewski, Isuru Ratnayake, Dinesh Pal Mudaranthakam.

**Project administration:** Prabhakar Chalise, Dinesh Pal Mudaranthakam.

**Resources:** Isuru Ratnayake.

**Supervision:** Prabhakar Chalise, Isuru Ratnayake, Dinesh Pal Mudaranthakam.

**Validation:** Stephanie Colwell.

**Writing – original draft:** Stephanie Colwell, Prabhakar Chalise, Byron Gajewski, Isuru Ratnayake, Dinesh Pal Mudaranthakam.

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
