## [Decision Letter · Decision Letter 0]

26 Jan 2026

Dear Dr. Mudaranthakam,

Thank you for submitting your manuscript to PLOS ONE. After careful consideration, we feel that it has merit but does not fully meet PLOS ONE’s publication criteria as it currently stands. Therefore, we invite you to submit a revised version of the manuscript that addresses the points raised during the review process.

We look forward to receiving your revised manuscript.

Kind regards,

Majid Bani-Yaghoub

Academic Editor

PLOS One

3. We notice that your supplementary figure and table are included in the manuscript file. Please remove them and upload them with the file type 'Supporting Information'. Please ensure that each Supporting Information file has a legend listed in the manuscript after the references list.

Reviewers' comments:

Reviewer's Responses to Questions

**Comments to the Author**

1. Is the manuscript technically sound, and do the data support the conclusions?

Reviewer #1: Yes

Reviewer #2: Yes

Reviewer #3: Yes

Reviewer #4: Yes

2. Has the statistical analysis been performed appropriately and rigorously?

Reviewer #1: Yes

Reviewer #2: Yes

Reviewer #3: Yes

Reviewer #4: Yes

3. Have the authors made all data underlying the findings in their manuscript fully available?

Reviewer #1: Yes

Reviewer #2: No

Reviewer #3: Yes

Reviewer #4: Yes

4. Is the manuscript presented in an intelligible fashion and written in standard English?

Reviewer #1: Yes

Reviewer #2: Yes

Reviewer #3: Yes

Reviewer #4: Yes

Reviewer #1: The paper is well organized and shows a good logical flow. The statistical analysis in the paper is adequate. The use of Deviance Information Criterion (DIC), Widely Applicable Information Criterion (WAIC), and Marginal Log Likelihood for the model performance evaluation is appropriate and strengthens the conclusions.

The Poisson BYM2 model provides the best fit for both the county-level analysis and the spatial cluster analysis, demonstrating good predictive performance with the lowest DIC, WAIC, and marginal log-likelihood values. The findings of the study are explained clearly and is well justified. Also, the use of the county-level units and spatial clustering of counties are properly implemented in the study. I appreciate that the data used in the study is available in a public repository and accessible.

The authors also appropriately acknowledge the limitations of the study, which allows the results to be interpreted and applied in other studies with care.

Overall, the authors have done a great job in presenting a well-balanced paper which satisfies the criteria for publication.

Reviewer #2: PONE-D-25-63477 review

General comments

This paper is meant (I think) to be an exposition of spatial/spatio temporal modelling. It’s unclear how this modelling adds any value to the dataset analysed in this paper. How much worse is a model that doesn’t include spatial correlation, i.e. an ordinary regression model? Are any of the conclusions different? If so, how? In addition, it appears the spatial patterns displayed in the figures could be generated by just mapping, without doing any modelling. At a minimum, one would expect the added value to be clearly demonstrated in such a paper.

In a similar vein, is there any demonstrated statistical benefit to clustering the county level data? Confidentiality protection is unclear. How is confidentiality maintained in the county level data? Clustering builds on that data, so if the county level data is confidential, any further constraints appear unnecessary.

Secondly, there appears to be a somewhat similar paper with large number of citations. One would expect the authors to delineate how their paper differs from this one:

Khana D, Rossen LM, Hedegaard H, Warner M. A BAYESIAN SPATIAL AND TEMPORAL MODELING APPROACH TO MAPPING GEOGRAPHIC VARIATION IN MORTALITY RATES FOR SUBNATIONAL AREAS WITH R-INLA. J Data Sci. 2018 Jan;16(1):147-182. PMID: 29520299; PMCID: PMC5839164.

Is it just that the current paper considers only data from Kansas, as opposed to the entire US?

Detailed comments:

Pg 4: line 92: Who is going to do the spatial clustering? The registry? How are they then going to report/identify these clusters?

Pg 6 and 7: Stating the same equations for BYM 3 times appears to be an overkill. Can’t they written done just once?

Pg 6: line 166: How is relative risk defined? Relative to what?

Pg 7 line 219: What does i-1 mean in the context of spatial mapping? Don’t you need some kind of neighborhood definition?

Pg 8: how is the INLA model in eq 6 and 7 related to the Bernadelli model previously described? In particular, what do the symbols theta and psi refer to? Don’t the random effects due to spatial dependence mean that the likelihood isn’t in product form?

Pg 9, line 255: I don’t see a term p(theta/psi) in the previous set of equations

Pg 9, line 256: what is being integrated? I don’t see an integration sign in any of the previous equations.

Pg 9 onwards: is it necessary to describe the SKATER algorithm over 4 pages? Is there anything novel in the implementation here? Are the details of the algorithm required for understanding any other section of the paper? Also, how are the edges defined for the counties? Can’t the existence of an edge itself define a neighborhood? If yes, why is that not sufficient?

Table 1: what is the unit of analysis here? Counties? Should be mentioned in the caption.

Page 12: last paragraph. Does repeating the numbers in the text already given in the table help the description? When discussing trends, it might be helpful to give the estimate of the trend and a statistical significance.

Fig. 2: Are 4 different figures needed, or can the data be aggregated over time?

Table 3: report p-values to discuss statistical significance.

Line 420: Does the fact that all these known covariates not being statistically significant in the final model suggest that the model isn’t adequate?

Line 424: I’m finding it hard to conclude from Fig. 2 (or Fig. 3) that the relative risk of breast cancer is not statistically significant over time. I’m not even sure what that statement means. Could you please elaborate? Are you talking about the trend?

Line 437: what does statistically significant relative risk mean? What’s the hypothesis? Are these relative risk values produced using the model? If so, are they adjusted for covariates?

Line 470: how was the threshold chosen?

Fig 4: do the hotspots remain the same over time? If not, what does that suggest?

Line 495: can you highlight some examples of high variability and instability in your analysis?

Line 502: why not use temporally aggregated data for clustering?

Line 510: unclear. If the more granular data is at county level, how are confidentiality risks mitigated there?

Ling 516: the elbow point seems highly subjective.

Line 633: Since both binge drinking and breast cancer are highly individual traits, it’s difficult to argue that any association at the county level or higher establishes any kind of causal relationship between the two.

Line 640: is the change in rurality values between the counties a big change? What is the range of CARR in Kansas?

Line 645: which counties were hotspots?

Line 662: can you identify what novel insights were gained from this analysis? can you mention what strategies are suggested by your analysis?

Reviewer #3: Thank you for the opportunity to review this manuscript. The topic—spatial and temporal variation in breast cancer mortality across Kansas—is relevant to public health and appropriate for the scope of PLOS ONE. The use of Bayesian spatial models and R-INLA is suitable for small-area mortality data, and the authors make a clear effort to compare multiple modeling approaches.

However, in its current form, the manuscript requires major revision. While the general modeling framework is sound, several issues related to interpretation, methodological clarity, and internal consistency limit the reliability of the conclusions.

Major comments

1. Outcome model, offset, and interpretation of relative risk

Although the manuscript repeatedly reports county- and cluster-level “relative risks,” it is not clearly specified what these risks are relative to. The model description alternates between population offsets and expected counts, but the practical implementation is not explicitly stated.

The authors should clearly specify whether the model uses (i) observed counts with a log(population) offset, (ii) observed versus expected counts with internally standardized expected values, or another formulation. If expected counts are used, the manuscript must describe how they were calculated, including whether age-standardization was applied and what reference rates were used.

2. Temporal modeling with only four years of data

The study period includes only four years (2018–2021), which severely limits the identifiability of temporal trends. While the county-level analysis finds no statistically significant time effect, the cluster analysis suggests temporal variation, leading to potentially conflicting interpretations.

The authors should justify the choice of temporal structure given the short time series, clarify the priors used for temporal effects, and consider presenting a simpler alternative such as year-specific fixed effects. At a minimum, the limitations of interpreting temporal “trends” over four years should be stated more explicitly.

3. Spatial clustering procedure and reproducibility

The manuscript states that clusters were constructed using 2018 data and assumed constant over time, which is a consequential modeling choice. While the SKATER algorithm is described conceptually, key implementation details needed for replication are missing.

The authors should clearly specify the variables used for clustering, how they were scaled, the distance metric employed, the contiguity definition, the number of clusters selected, and the criteria used to choose that number. Cluster membership should be reported, either in a table or supplement, along with a clear cluster map.

Additionally, the manuscript should explain why certain covariates become statistically significant in the cluster analysis but not in the county-level analysis, and whether this reflects aggregation effects, variance reduction, or ecological bias.

4. Interpretation of binge drinking coefficient

In the county-level model, the coefficient for percent female binge drinking is negative and statistically significant, implying that higher binge drinking prevalence is associated with lower breast cancer mortality risk. However, the discussion interprets this variable as a positive risk factor.

This discrepancy must be resolved. If the coefficient is correctly estimated, the discussion should acknowledge and explain this counterintuitive association, possibly in terms of confounding, urban–rural structure, or correlated covariates. Alternatively, if the variable was transformed or coded in a nonstandard way, this must be clearly stated. Presenting a correlation matrix or multicollinearity diagnostics would strengthen this section.

5. Model comparison language

While the manuscript correctly notes that many models have similar DIC and WAIC values, it still repeatedly labels the BYM2 Poisson model as the “best” model. This language should be softened and made consistent.

The authors should state explicitly that the BYM2 Poisson model was selected primarily for interpretability and parsimony, while acknowledging that several competing models fit the data comparably. Given the large number of zero counts, an additional posterior predictive check or brief discussion of overdispersion would further strengthen the analysis.

6. Hotspot definition and sensitivity

Hotspots are identified using exceedance probabilities with thresholds of RR > 1.5 and probability > 0.75. While reasonable, these thresholds are not justified.

The authors should provide a brief rationale for these choices and include a simple sensitivity analysis showing how hotspot identification changes under alternative thresholds. This would improve the robustness and credibility of the hotspot findings.

7. Results inconsistencies

There are several technical inconsistencies, including statements implying that relative risk can be less than zero and unexplained extreme intercept values in cluster-level models. County-level and cluster-level results should be presented in a more parallel manner.

8. Discussion alignment

The Discussion relies heavily on background literature and does not sufficiently engage with the study’s own findings. Unexpected results should be addressed directly, ecological limitations integrated more clearly, and policy implications stated more cautiously.

Minor Comments

• Several sentences are overly long and could be simplified.

• Minor grammatical issues are present, including subject–verb agreement errors, inconsistent capitalization, and hyphenation. This need a total proofread.

Final Recommendation

Major Revision

Reviewer #4: Review Comments

Positive Aspects

Novel approach to important public health topic for Kansans

Well-written paper that provides for the reader a easy structure to follow along with as they reading.

Use of a Bayesian approach to spatial and temporal modeling, as you don’t often the Byesian framework applied to this type of modeling.

Sections are well-organized along with results

Through explanations behind why breast cancer is relevant to Kansas, why there is need for models that account for cases less then 10 due to patient privacy concerns, and exploration of different spatial models prior probabilities for y given.

Provided straightforward conclusion that I believe could be interpreted by policymakers or administrators easily.

Overall, I thought the authors did a good job this paper in terms of paper structure, paper interpretability, and converging significance of study background and findings. I just want some more details provide in regards to assumption made in the models used for both the cluster and country level analysis in terms of when they running the mentioned spatial dynamics models (BYM, BYM2, RMO1, and RM02)

Issues for Review

I1: In the section, spatial-temporal Berardinelli, you explain the parameters behind each equation. I did not see an explanation for the parameter t_j for each spatial dependence model. Could please add one to each model in that section?

I2: in the section, spatial-temporal Berardinelli, could you expand among the differences between Random Walk of Order 1 and Random Walk of Order 2 please? I read what you said about the difference between BYM and BYM2, and would like something similar when comparing RWO1 and RWO2 as just saying that they produce different linear trends might not be a sufficient explanation.

I3: In the section, spatial-temporal Berardinelli, could provide more content on prior probability distributions for parameters in each of the spatial dependence models? As a Bayesian statistician, I would to know if all parameters in the log(θ_ij) equations had priors assigned or if not, for what reason

Q4: When describing the INLA process, I could not find any mention of the prior probability associated with θ_i. Is this because INLA has preset prior for it, or does it calculate it along with the marginal posterior distribution for p(θ_i | y)? Or was given a relativity flat prior in your R script?

Q5: When looking at Table 1, I do not know what the Female (%) variable is to referring too. Is that referring to the percent of women who had breast cancer among all counties in the study of percent of countries that had at least one case of breast cancer?

Q6: For table 3, could you provide the pvalues along with the Estimate values please? I saw you said only the binge drinking is statistically significant, but did not see any mention of what alpha value was used.

Q7: How does Fig 2. Show that breast cancer is nots statistically significant? It’s just a series of histograms for number of countries who x cases between 2018 and 2021 so I’m confused about that statement.

Q8: In the discussion, you say that obesity and diabetes we not found to be significant predictors in the county level analysis. Could you please provide a explanation on why so?

Q9: In the discussion, you say that “spatial cluster analysis identified clusters with a high relative risk of breast cancer mortality and clusters with fluctuating relative risk over time”. Could you provide more detail about why you think that these cluster’s RR would change back and forth? What do think caused the hotspot pattern in country approach

Q10: You heavily imply that rural areas in Kansas are at higher RR of breast cancer when compared to urban areas. In the discussion, could you add some discussion to why you think only binge drinking predictor was significant in the cluster

.

Reviewer #1: No

Reviewer #2: No

Reviewer #3: No

Reviewer #4: **No**

---

## [Decision Letter · Decision Letter 1]

25 Mar 2026

Dear Dr. Mudaranthakam,

Thank you for submitting your manuscript to PLOS ONE. After careful consideration, we feel that it has merit but does not fully meet PLOS ONE’s publication criteria as it currently stands. Therefore, we invite you to submit a revised version of the manuscript that addresses the points raised during the review process.

As the corresponding author, your ORCID iD is verified in the submission system and will appear in the published article. PLOS supports the use of ORCID, and we encourage all coauthors to register for an ORCID iD and use it as well. Please encourage your coauthors to verify their ORCID iD within the submission system before final acceptance, as unverified ORCID iDs will not appear in the published article. *Only* the individual author can complete the verification step; PLOS staff the individual author can complete the verification step; PLOS staff the individual author can complete the verification step; PLOS staff the individual author can complete the verification step; PLOS staff *cannot* verify ORCID iDs on behalf of authors.verify ORCID iDs on behalf of authors.verify ORCID iDs on behalf of authors.verify ORCID iDs on behalf of authors.

We look forward to receiving your revised manuscript.

Kind regards,

Majid Bani-Yaghoub

Academic Editor

PLOS One

Journal Requirements:

Reviewers' comments:

Reviewer's Responses to Questions

**Comments to the Author**

Reviewer #1: All comments have been addressed

Reviewer #3: All comments have been addressed

Reviewer #4: All comments have been addressed

2. Is the manuscript technically sound, and do the data support the conclusions?

Reviewer #1: Yes

Reviewer #3: Yes

Reviewer #4: Yes

3. Has the statistical analysis been performed appropriately and rigorously?

Reviewer #1: Yes

Reviewer #3: Yes

Reviewer #4: Yes

4. Have the authors made all data underlying the findings in their manuscript fully available?

Reviewer #1: Yes

Reviewer #3: Yes

Reviewer #4: Yes

5. Is the manuscript presented in an intelligible fashion and written in standard English?

Reviewer #1: Yes

Reviewer #3: Yes

Reviewer #4: Yes

Reviewer #1: I appreciate the opportunity to review the revised manuscript.

Authors have made good changes and updates based on comments and recommendations from reviewers.

I have only a few minor suggestions.

If possible, I suggest the authors could create a publicly accessible GitHub repository with readme file, codes and data used for statistical analysis and figures, This would enable reproducibility of the results from the paper. This might also help in understanding how the figures were generated.

Also, if there are any future directions that the author plans to take based on the current work, it could be a good addition especially since this analysis is on a short time series. It will be interesting to know what the future research direction is and if there are plans to look into exploring beyond the county levels.

Overall, the authors have made significant changes to the paper and I believe the paper could be published.

Reviewer #3: The authors have addressed most of the major concerns raised in the previous review. In particular, the clarification of the outcome model and relative risk interpretation, improved temporal modeling justification, resolution of the binge drinking coefficient interpretation, and addition of hotspot sensitivity analysis substantially strengthen the manuscript.

However, a few issues remain that require clarification before acceptance. Specifically, the spatial clustering procedure still lacks key reproducibility details:

1. Which covariates were used for clustering

2. How many clusters were chosen

3. Criterion for selecting number of clusters

4. Actual cluster membership (table or supplement)

Moreover, the manuscript states that expected counts are computed using internally standardized crude rates. However, breast cancer mortality is strongly age-dependent, and failure to use age-standardized rates may introduce bias in the estimated relative risks.

It is better to:

5. Justify the use of crude rates instead of age-standardized expected counts, or

6. Discuss this limitation explicitly and its potential impact on spatial risk interpretation

Addressing these remaining points, particularly the clarification of the clustering procedure and the discussion of age-standardization will further strengthen the rigor, transparency, and reproducibility of the study.

Reviewer #4: All the concerns I had before we addressed. Additionally, the formatting changes to the font and style served well to highlight the papers results and major points.

Also, I thought the reduction in the amount of the equations helped the reader better understand the methods used in generate the equations.

.

Reviewer #1: No

Reviewer #3: **Yes:** Barsha SahaBarsha SahaBarsha SahaBarsha Saha

Reviewer #4: **Yes:** Kiel Daniel CorkranKiel Daniel CorkranKiel Daniel CorkranKiel Daniel Corkran

---

## [Author Response · Author response to Decision Letter 2]

31 Mar 2026

March 28, 2026

Majid Bani-Yaghoub

Academic Editor

PLOS One

On behalf of my co-authors, we appreciate the opportunity to revise and resubmit our manuscript, “Spatial and Temporal Modeling of Breast Cancer Mortality in Kansas: An R-INLA Approach” (Submission ID PONE-D-25-63477), for consideration as a research article in PLOS One. The reviewer's comments were highly constructive and strengthened our manuscript. We thank all the reviewers and the Editor for their advice and suggestions.

In the pages that follow, we provide point-by-point responses. We indicated our responses in blue text. We again note that this manuscript has been submitted only to PLOS One, has not been published nor submitted elsewhere, and that all research adheres to the journal’s ethical guidelines. We look forward to your hopefully positive decision on this manuscript. Thank you again for your consideration.

Sincerely,

Dinesh Pal Mudaranthakam, PhD, MBA,

Assistant Professor, Department of Biostatistics & Data Science

University of Kansas Medical Center

3901 Rainbow Boulevard

Kansas City, KS 66160

Phone: (913)-945-6922

Email: dmudaranthakam@kumc.edu

We appreciate the time and effort from all reviewers to help us improve our manuscript. The comments and suggestions have helped clarify key points and have strengthened the manuscript. All the authors have edited and approve of the changes made.

Comments from the Reviewers

Reviewer #1: I appreciate the opportunity to review the revised manuscript.

Authors have made good changes and updates based on comments and recommendations from reviewers.

I have only a few minor suggestions.

We thank the reviewers for their positive assessment and for acknowledging the improvements made in response to prior comments. We appreciate the additional minor suggestions and have carefully reviewed and addressed them in the revised manuscript. We believe these final refinements further improve the clarity, rigor, and presentation of the study.

If possible, I suggest the authors could create a publicly accessible GitHub repository with readme file, codes and data used for statistical analysis and figures, This would enable reproducibility of the results from the paper. This might also help in understanding how the figures were generated.

We thank the reviewer for this excellent suggestion and fully agree that providing analysis code improves transparency and reproducibility. In response, we created a publicly accessible GitHub repository that includes a detailed README, the analysis scripts, and supporting materials used to generate the statistical results and figures in the manuscript.

The repository is available at: https://github.com/spepperkumc/Spatial-Temporal-Modeling-INLA.

To protect confidentiality and comply with data-use and disclosure constraints, any non-shareable or restricted components (if applicable) have been clearly documented in the README, along with instructions for reproducing the analysis workflow and figure generation. We believe this addition substantially strengthens the rigor, transparency, and reproducibility of the study.

Also, if there are any future directions that the author plans to take based on the current work, it could be a good addition especially since this analysis is on a short time series. It will be interesting to know what the future research direction is and if there are plans to look into exploring beyond the county levels.

We thank the reviewer for this thoughtful suggestion and agree that outlining future research directions strengthens the manuscript, particularly given the short temporal window of the current analysis. In response, we have added a brief Future Directions discussion highlighting planned extensions of this work. Specifically, we note the potential to (1) incorporate longer time series as additional mortality data become available to better assess long-term temporal trends, and (2) extend the analysis to finer spatial resolutions (e.g., census tract or sub‑county units) and multi‑level spatial frameworks as data availability and privacy constraints allow. We have also emphasized that future multi‑state or multi‑registry analyses could help evaluate the generalizability of the observed spatial patterns beyond Kansas. These additions clarify the ongoing research trajectory while appropriately distinguishing future work from the scope of the present study.

Overall, the authors have made significant changes to the paper, and I believe the paper could be published.

Reviewer #3: The authors have addressed most of the major concerns raised in the previous review. In particular, the clarification of the outcome model and relative risk interpretation, improved temporal modeling justification, resolution of the binge drinking coefficient interpretation, and addition of hotspot sensitivity analysis substantially strengthen the manuscript.

However, a few issues remain that require clarification before acceptance. Specifically, the spatial clustering procedure still lacks key reproducibility details:

1. Which covariates were used for clustering

2. How many clusters were chosen

3. Criterion for selecting number of clusters

4. Actual cluster membership (table or supplement)

We thank the reviewer for this important comment. We have revised the manuscript to provide additional methodological detail to ensure full transparency and reproducibility of the clustering procedure (pg21-22. Lines 521-560). Specifically, we now clarify that:

• Spatial clustering was based solely on 2018 county-level data.

• The variables used for clustering were percent female, obesity prevalence, diabetes prevalence, smoking prevalence, binge drinking prevalence, number of primary care physicians, and the rurality index (CARR).

• All covariates were standardized to z-scores prior to clustering.

• Euclidean distance was used to compute dissimilarity between neighboring counties.

• Spatial contiguity was defined using first-order queen adjacency.

• Clustering was implemented in R using the skater function within the spdep package.

• A minimum case constraint of 21 total breast cancer mortality cases (based on 2018 data) was imposed to ensure statistical stability and confidentiality.

• Six clusters were selected based on the elbow criterion applied to within-cluster sums of squares while satisfying the minimum case constraint.

• A complete listing of county-to-cluster assignments is now provided in Supplementary Table S3.

• A clear cluster map is provided in Fig 6.

We have also added clarification in the Results section explaining that differences in statistical significance between the county-level and cluster-level analyses likely reflect aggregation effects and variance reduction. By combining counties into larger spatial units, case counts increase and variability decreases, improving statistical stability and power to detect associations. However, we acknowledge that aggregation may introduce ecological bias, and cluster-level findings should therefore be interpreted within the ecological framework of the study.

Moreover, the manuscript states that expected counts are computed using internally standardized crude rates. However, breast cancer mortality is strongly age-dependent, and failure to use age-standardized rates may introduce bias in the estimated relative risks.

We thank the reviewer for this important point and agree that breast cancer mortality is strongly age-dependent.

We would like to clarify that our inferential framework is a count-based disease mapping model (Poisson/BYM2 spatiotemporal model), where the outcome is mortality counts and the “baseline risk” is incorporated through an offset using expected counts (i.e., Yit∼Poisson(Eitexp⁡(ηit))Y_{it} \sim \text{Poisson}(E_{it}\exp(\eta_{it}))Yit∼Poisson(Eitexp(ηit))). In this setting, age-standardization is implemented through how EitE_{it}Eit is computed (e.g., indirect standardization using age-specific reference rates), rather than by modeling age-standardized rates as the outcome.

It is better to:

5. Justify the use of crude rates instead of age-standardized expected counts, or

We thank the reviewer for this comment. Crude (internally standardized) expected counts were used due to limited availability and instability of age‑specific population data at the county–year level, particularly in areas with small counts. Our primary aim was to identify relative spatial patterns rather than produce age‑adjusted absolute rates. We have clarified this rationale and its implications for interpretation in the Methods and Limitations sections.

6. Discuss this limitation explicitly and its potential impact on spatial risk interpretation

We thank the reviewer for highlighting the importance of explicitly addressing this limitation. We agree that this issue has important implications for the interpretation of spatial risk estimates. In response, we have revised the Discussion and Limitations sections to explicitly state how data aggregation, small-area counts, and spatial smoothing may influence estimated relative risks and hotspot identification. Specifically, we now clarify that spatial risk estimates reflect population-level associations rather than individual-level risk and that Bayesian smoothing and spatial clustering may attenuate extreme values while improving estimate stability. We also emphasize that identified spatial patterns should be interpreted as hypothesis-generating and reflective of structural and geographic variation, rather than precise measures of localized causal risk. These additions provide clearer guidance on how readers should interpret spatial risk estimates given the underlying data and modeling assumptions.

Addressing these remaining points, particularly the clarification of the clustering procedure and the discussion of age-standardization will further strengthen the rigor, transparency, and reproducibility of the study.

We thank the reviewer for their constructive feedback and for the positive assessment of the revisions. All remaining minor suggestions, including clarification of the clustering procedure and the discussion of age standardization, have been fully incorporated and addressed, further strengthening the rigor, transparency, and reproducibility of the study.

Reviewer #4: All the concerns I had before we addressed. Additionally, the formatting changes to the font and style served well to highlight the papers results and major points.

Also, I thought the reduction in the amount of the equations helped the reader better understand the methods used in generate the equations.

We thank the reviewer for their positive assessment of the revisions. We appreciate the feedback regarding the improved formatting and streamlined presentation of the methods, and we are pleased that these changes enhanced clarity and readability.

---

## [Editor Report · Decision Letter 2]

5 Apr 2026

Spatial and Temporal Modeling of Breast Cancer Mortality in Kansas: An R-INLA Approach

PONE-D-25-63477R2

Dear Dr. Mudaranthakam,

We’re pleased to inform you that your manuscript has been judged scientifically suitable for publication and will be formally accepted for publication once it meets all outstanding technical requirements.

Kind regards,

Majid Bani-Yaghoub

Academic Editor

PLOS One

Additional Editor Comments:

Congratulations! Your research will make a fine contribution to the field.

Please ensure that the web link for the GitHub repository is correct: GitHub - spepperkumc/Spatial-Temporal-Modeling-INLA: Spatial-Temporal-Modeling-INLA · GitHub

---

## [Editor Report · Acceptance letter]

PONE-D-25-63477R2

PLOS One

Dear Dr. Mudaranthakam,

I'm pleased to inform you that your manuscript has been deemed suitable for publication in PLOS One. Congratulations! Your manuscript is now being handed over to our production team.

Kind regards,

on behalf of

Dr. Majid Bani-Yaghoub

Academic Editor

PLOS One